# BAG6 inhibits influenza A virus replication by inducing viral polymerase subunit PB2 degradation and perturbing RdRp complex assembly

Yong Zhou[1,2,3], Tian Li[1,2,3], Yunfan Zhang[1,4], Nianzhi Zhang[1,3], Yuxin Guo[1,2,3], Xiaoyi Gao[1,2,3], Wenjing Peng[1,2,3], Sicheng Shu[1,2,3], Chuankuo Zhao[1,2,3], Di Cui[1,4], Honglei Sun[1,2,3], Yipeng Sun[1,2,3], Jinhua Liu[1,2,3], Jun Tang[1,4], Rui Zhang[1,4]*, Juan Pu[1,2,3]*

1 National Key Laboratory of Veterinary Public Health and Safety, College of Veterinary Medicine, China Agricultural University, Beijing, China, 2 Key Laboratory for Prevention and Control of Avian Influenza and Other Major Poultry Diseases, Ministry of Agriculture and Rural Affairs, College of Veterinary Medicine, China Agricultural University, Beijing, China, 3 Department of Preventive Veterinary, College of Veterinary Medicine, China Agricultural University, Beijing, China, 4 Department of Basic Veterinary, College of Veterinary Medicine, China Agricultural University, Beijing, China

* 2015001@cau.edu.cn (RZ); pujuan@cau.edu.cn (JP)

**Data Availability Statement:** All the values behind the means and used to build graphs have been deposited to public repository at https://doi.org/10.

## Abstract

The interaction between influenza A virus (IAV) and host proteins is an important process that greatly influences viral replication and pathogenicity. PB2 protein is a subunit of viral ribonucleoprotein (vRNP) complex playing distinct roles in viral transcription and replication. BAG6 (BCL2-associated athanogene 6) as a multifunctional host protein participates in physiological and pathological processes. Here, we identify BAG6 as a new restriction factor for IAV replication through targeting PB2. For both avian and human influenza viruses, overexpression of BAG6 reduced viral protein expression and virus titers, whereas deletion of BAG6 significantly enhanced virus replication. Moreover, BAG6-knockdown mice developed more severe clinical symptoms and higher viral loads upon IAV infection. Mechanistically, BAG6 restricted IAV transcription and replication by inhibiting the activity of viral RNA-dependent RNA polymerase (RdRp). The co-immunoprecipitation assays showed BAG6 specifically interacted with the N-terminus of PB2 and competed with PB1 for RdRp complex assembly. The ubiquitination assay indicated that BAG6 promoted PB2 ubiquitination at K189 residue and targeted PB2 for K48-linked ubiquitination degradation. The antiviral effect of BAG6 necessitated its N-terminal region containing a ubiquitin-like (UBL) domain (17-92aa) and a PB2-binding domain (124-186aa), which are synergistically responsible for viral polymerase subunit PB2 degradation and perturbing RdRp complex assembly. These findings unravel a novel antiviral mechanism via the interaction of viral PB2 and host protein BAG6 during avian or human influenza virus infection and highlight a potential application of BAG6 for antiviral drug development.

6084/m9.figshare.25152692. All the raw images of
WB and IF are uploaded as Supporting information.

**Funding:** This work was supported by the National
Key Research and Development Program of China
(2021YFD1800202 to JP), the National Natural
Science Foundation of China (32172829 to RZ and
32192450 to JL). The funders had no role in study
design, data collection and analysis, decision to
publish, or preparation of the manuscript.

**Competing interests:** The authors have declared
that no competing interests exist.

## Author summary

Influenza A virus (IAV) is a major public health threat worldwide. The viral polymerase subunit PB2 of IAV is not only responsible for transcription and replication of viral RNA but also acts as a key determinant for host adaptation, making it a key target of host defense system. In this study, we report BAG6 as a novel negative regulator of viral PB2 protein. It interacts with the N-terminus of PB2 and promotes PB2 ubiquitination at the K189 residue, which is highly conserved in all IAV subtypes, resulting in the competitive inhibition of RdRp assembly and the ubiquitination degradation of PB2. These findings emphasize the potent antiviral activity of BAG6 against both avian and human influenza viruses and provide a detailed insight to facilitate antivirals targeting PB2 protein.

## Introduction

Influenza A virus (IAV) poses a significant threat to public health globally, causing seasonal epidemics and at least 500,000 deaths each year [1–3]. H1N1 and H3N2 are the predominant strains of human IAV responsible for yearly epidemics. Additionally, increasing avian-origin IAV strains, such as H5N1, H7N9 and H9N2, have acquired the ability to cross the species barrier to infect humans and cause high mortality rates of up to 50% [4–7].

IAV is enveloped negative-strand RNA viruses that contain 8 genome segments. The transcription and replication of IAV genome are catalyzed by the viral ribonucleoprotein (vRNP) complex composed of an RNA-dependent RNA polymerase (RdRp) and multiple copies of viral nucleoproteins (NP). The RdRp is a heterotrimeric complex containing three subunits: polymerase basic protein 2 (PB2), polymerase basic protein 1 (PB1), and polymerase acidic protein (PA). The three subunits play distinct roles within the polymerase, and are all essential for viral transcription and replication. Once an RdRp complex moves into the host cell nucleus, it initiates viral mRNA transcription by a 'cap-snatching' mechanism[8–10]. This process requires cleaving an mRNA cap-containing oligonucleotide from host cell pre-mRNA, and extending it using viral genomic RNA as a template. The cap-binding and endonuclease activities are activated by the signalling from the RNA-binding PB1 subunit to the cap-binding PB2 subunit, and the interface between these two subunits is essential for the polymerase activity. In addition, the viral polymerase subunits especially PB2 can influence the host range of IAVs. A specific E627K mutation in PB2 has been identified as the dominant host-adaptive mutation in most human-adapted IAVs, enable the avian-origin influenza viruses to overcome restriction barriers and replicate efficiently in human cells [11–14]. In turn, this makes the polymerase subunits interesting targets for host defense system and drug design [15].

BAG6 (BCL2-associated athanogene 6, also known as BAT3 or Scythe) is a member of the BAG family due to a sequence homologous to the BAG domain at its C terminus and the collaboration with Heat Shock Protein 70 (Hsp70) family molecular chaperones [16]. BAG6 was originally identified as a gene located in the human major histocompatibility complex (MHC) III locus on chromosome 6 that contains many genes essential for immune functions [17,18]. Subsequent studies discovered BAG6 participates in a variety of physiological and pathological processes, including the protein quality control [16,19], T cell response [20], and apoptosis [21–23]. In addition, the expression of BAG6 exhibits a relatively higher levels in some tissues of multi-cellular organisms, such as testis and brain [24], suggesting its potential function in spermatogenesis and brain development. Due to the diversity of its functions, the significance of BAG6 in disease and therapy is of particular interest. However, the role of BAG6 in virus infection has not been explored.

In this study, we identified BAG6 as an interaction partner of PB2 and thus restrict IAV replication *in vitro* and *in vivo*. BAG6 can bind to the N-terminus of viral PB2 polymerase sub-unit to perturb the RdRp complex formation and target PB2 for K48-linked ubiquitination degradation, thereby inhibiting viral polymerase activity and restricting IAV replication. These findings reveal a critical mechanism allowing the fast tracking of viral polymerase sub-unit PB2 and highlight the potential of using BAG6 for the development of novel anti-IAV therapeutics.

## Results

### The overexpression of BAG6 inhibits IAV replication

To clarify the role of BAG6 in influenza virus replication, we transfected the BAG6-HA expression plasmid into A549 cells or HeLa cells, and then infected the cells with the IAV human strain H1N1 (PR8) at an MOI of 1. At different time points post-infection, the viral proteins expressed in cells and the amount of virus released to the culture supernatant were examined by western blotting and $TCID_{50}$ assay, respectively. The results showed that the expression of viral proteins (NP and M1) was significantly reduced in BAG6-overexpressing (BAG6-HA) A549 cells (Fig 1A) and HeLa cells (S1A Fig) compared with that in empty vector (EV) control cells. The virus titers, as determined by $TCID_{50}$ assay, was 10- to 15-fold lower in BAG6-over-expressing cells than that in control cells (Figs 1B and S1B), suggesting that BAG6 suppresses H1N1 replication. To determine if BAG6 also has antiviral activity against other subtypes of influenza viruses, we observed the viral protein and virus titers in A549 cells infected with avian H7N9 (CN0606), H9N2 (SD196) or human H5N1 (AH1) strains. Consistently, the ectopic expression of BAG6 also caused a significant decrease in viral protein expression (Fig 1C, 1E and, 1G) and an up to 15-fold reduction in viral loads (Fig 1D, 1F, and 1H) of H7N9, H9N2 and H5N1.

### Knockout of BAG6 enhances IAV replication

To further confirm the antiviral activity of BAG6 against IAV, we then generated a BAG6 knockout (BAG6-KO) A549 cell line using CRISPR/Cas9 gene editing system, and verified that the knockout of BAG6 did not affect cell growth viability by CCK-8 assay (S2 Fig). We then infected BAG6-KO or WT A549 cells with the H1N1 (PR8), and examined whether BAG6 deficiency affected IAV replication. As shown in Fig 2, an increased NP protein level (Fig 2A) and an up to 10-fold higher viral titers (Fig 2B) were observed in BAG6-KO cells than in BAG-WT cells. The consistent results were also observed in BAG6 knockout (BAG6-KO) HeLa cell line infected with H1N1 (S1C and S1D Fig), as well as in BAG6-KO A549 cells infected with other subtypes of influenza viruses, including avian A/H7N9 (CN0606), A/H9N2 (SD196) and human A/H5N1 (AH1) strains (Fig 2C and 2D), compared with the correspond-ing BAG6-WT cells. Next, to further confirm that the observed enhancement of IAV replica-tion was indeed caused by specific depletion of BAG6 but not by off-target effects, we determined whether the phenotype could be rescued by a BAG6-HA expression plasmid in BAG6-KO A549 cells. The viral NP expression (Fig 2E) and virus titers (Fig 2F) up-regulated by BAG6-knockout were restored by BAG6-HA expression in BAG6-KO A549 cells infected with H1N1. Due to the potent antiviral activity of BAG6, we next determined whether IAV was able to regulate BAG6 expression during virus infection to evade its antiviral effects. The results showed that the transcription and expression of BAG6 was significantly suppressed by different subtypes of IAV strains (S3 Fig). Collectively, these findings suggest that BAG6 plays an important role in inhibiting the replication of different influenza virus subtypes from avian and human.

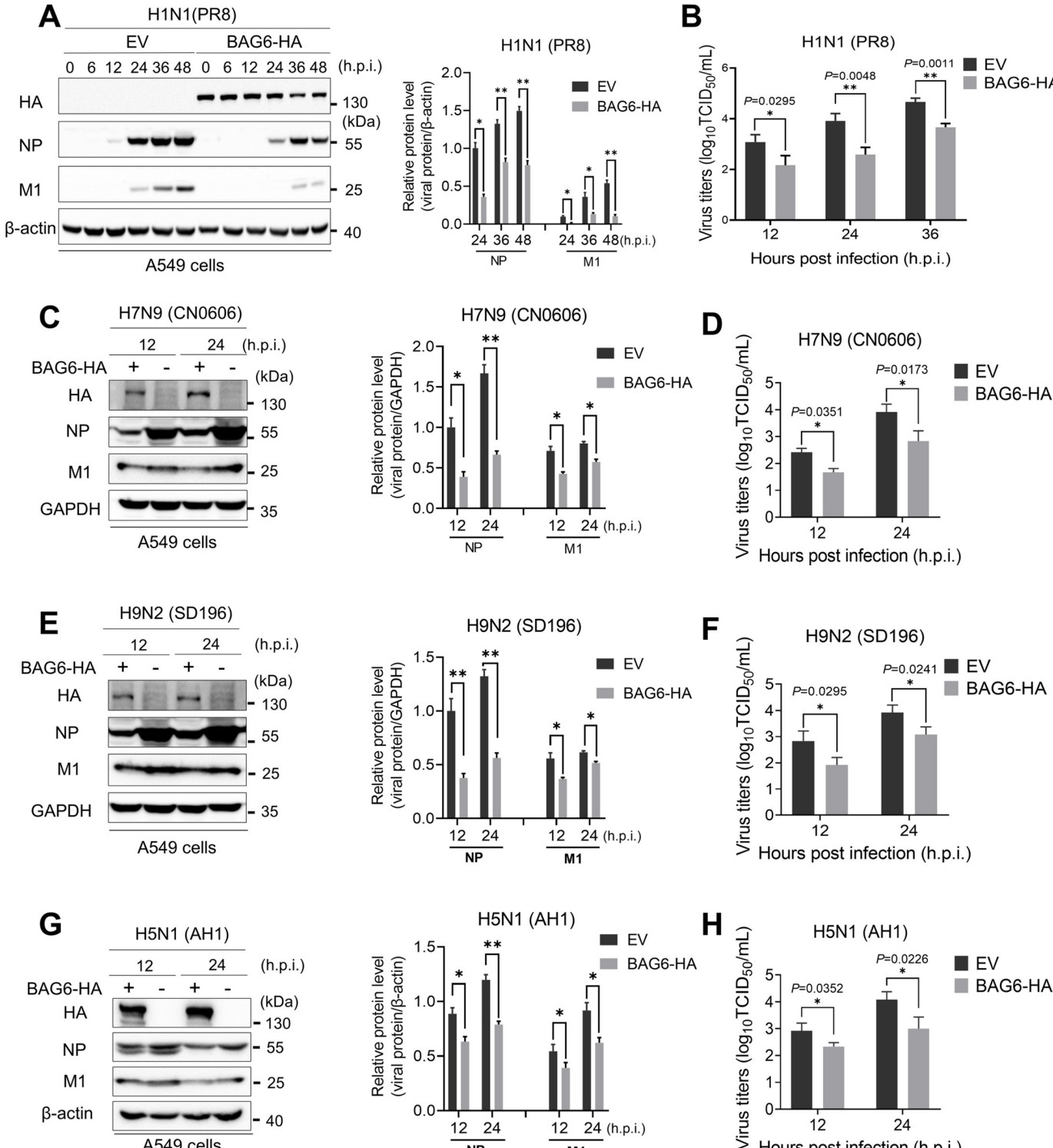

**Fig 1. The overexpression of BAG6 inhibits IAV replication.** (A and B) A549 cells were transfected with BAG6-HA or Empty vector (EV) plasmids. At 24 h post-transfection, the cells were infected with IAV H1N1 (MOI = 1.0). The viral NP and M1 expression were measured by western blotting at different time points postinfection, as indicated (A, left panels), and densitometry analysis and quantification were performed (A, right panels). Viral titers in the supernatants were determined by $TCID_{50}$ assay at 12, 24 and 36 h post-infection (B). (C-H) A549 cells were transfected with BAG6-HA or Empty vector (EV) plasmids. At 24 h post-transfection, the cells were infected with IAV H7N9 (C), H9N2 (E) and H5N1 (G) (MOI = 1.0). The viral NP expression was measured by western blotting at 12 and 24 h post-infection, as indicated (left panels), and densitometry analysis and quantification were performed (right panels). Viral titers in the supernatants were

determined by TCID$_{50}$ assay at 12 and 24 h post-infection with H7N9 (D), H9N2 (F) and H5N1 (H). Data presented as means ± SD and are representative of three independent experiments. *$p < 0.05$, **$p < 0.01$, ***$p < 0.001$, Unpaired Student's t test.

## BAG6 inhibits IAV replication *in vivo*

To investigate the role of BAG6 during IAV infection *in vivo*, BAG6-knockdown mice were generated using a peptide-conjugated phosphorodiamidate morpholino oligomers (PPMOs) before PR8 virus (100 TCID$_{50}$) or mock infection (Fig 3A). PPMOs are sequence-specific anti-sense agents that can be administered intranasally to elicit a transient reduction in the expression of the targeted gene product in the lungs of treated mice [25–28]. The successful depletion of BAG6 in lung by BAG6-targeting PPMO (PPMO-BAG6) was confirmed by western blotting compared with PPMO control (PPMO-NC) before and after virus infection (Fig 3B). The body weight and survival monitoring experiments showed that knocking down BAG6 alone (PPMO-BAG6-Mock) did not affect the weight and survival rate of mice (Fig 3C and 3D). However, the BAG6-knockdown mice (PPMO-BAG6-PR8) exhibited a more significant weight loss after IAV infection compared to the mice treated with PBS (PBS-PR8) or PPMO-NC (PPMO-NC-PR8) (Fig 3C). Moreover, while 40% of the PPMO-BAG6-PR8 mice succumbed to infection at 8 dpi, 100% of the PBS-PR8 or PPMO-NC-PR8-treated mice survived after infection (Fig 3D). Furthermore, the virus loads in the lungs of PPMO-BAG6-PR8 mice were up to approximate 10-fold higher than that in control mice at 3 and 5 day post infection (d.p.i.) (Fig 3E), which was consistent with the immunohistochemical staining of viral protein NP showing an increased viral antigen level in PPMO-BAG6-PR8 mice (Fig 3F). The histopathological analysis revealed more severe injury and inflammation in lung tissue of PPMO-BAG6-PR8 mice at 3 and 5 d.p.i. compared to those of control mice (Fig 3G). These results demonstrate that BAG6 plays a crucial role in inhibiting IAV replication in mice.

## BAG6 inhibits IAV replication independent of IFN-mediated innate immune pathways

Type I Interferon (IFN-I) plays a critical role in defending against viral infection and regulating innate immune response. A recent study reported that BAG6 was capable of regulating SeV-induced activation of IFN-I by inhibiting RIG-I/VISA-mediated innate immune response [29]. To assess the role of IFN-I pathway in BAG6-mediated anti-IAV response, we firstly examined the expressions of several key components involved in IFN-I signaling pathways, including pattern recognition receptor RIG-I, the phosphorylated STAT1 (P-STAT1) and the IFN-stimulated gene ISG15, in both BAG6 overexpressing and knocking-out HeLa cells during PR8 infection. The results showed that the expressions of RIG-I, P-STAT1 and ISG15 were only slightly increased in BAG6-overexpressing cells (Fig 4A), and exhibited a comparatively obvious decrease in BAG6-deficient cells (Fig 4B). To further verify whether BAG6 inhibits IAV replication through IFN-I innate immune pathways, we used an IRF9-knockout HeLa cell line (HeLa-IRF9-KO), in which the IFN-I-mediated innate immune responses were disrupted by knocking out interferon regulation factor 9 (IRF9) in JAK-STAT pathway [30,31], to examine the effect of BAG6-HA on IAV replication. Interestingly, while the depletion of IRF9 completely inhibited the expression of IFN-I-induced antiviral factor ISG15 in PR8-infected cells, BAG6-HA still remained its ability to suppress the expression of viral NP protein (Fig 4C) and induced an up to 10-fold reduction in the virus titer (Fig 4D), which are equal to the reduced levels in HeLa-WT cells. In addition, we also examined the inhibitory effect of BAG6 on PR8 replication in Vero cell line, since it has been reported as a cell line with a deficient

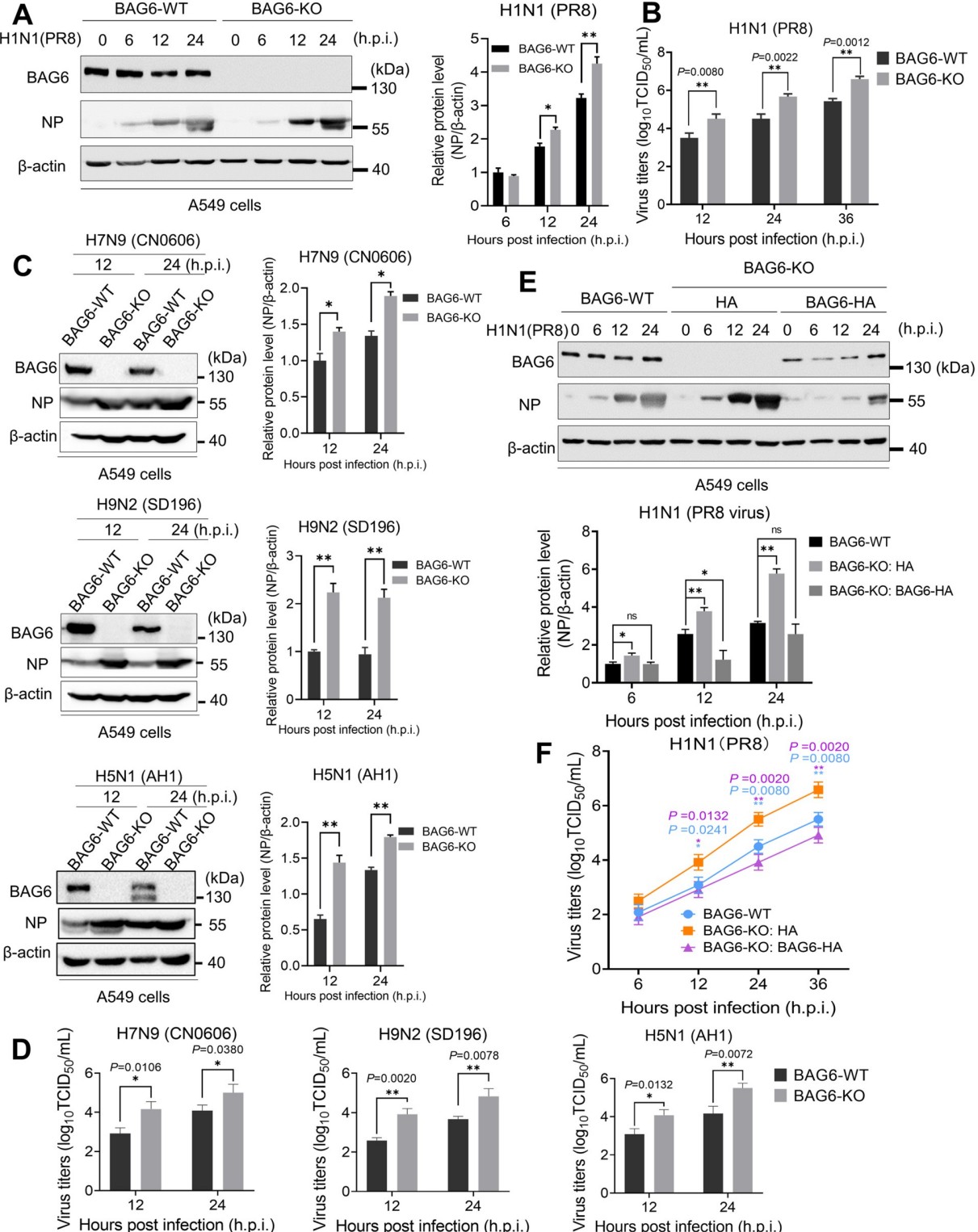

**Fig 2. BAG6 knockout enhances IAV replication.** (A and B) BAG6-KO and BAG6-WT A549 cells were infected with IAV H1N1 (MOI = 1.0), and viral NP expression were measured by western blotting at different time points post-infection, as indicated (A, left panels), and densitometry analysis and quantification were performed (A, right panels). Viral titers in the supernatants were determined by TCID$_{50}$ assay at 12, 24 and 36 h post-infection (B). (C and D) BAG6-KO and BAG6-WT A549 cells were infected with IAV H7N9 or H9N2 or H5N1 (MOI = 1.0), and the NP expression were measured by western blotting at different time points post-infection (C, left panels), and densitometry

analysis and quantification were performed (C, right panels). Viral titers in the supernatants were determined by $TCID_{50}$ assay at 12 and 24 h post-infection (D). (E and F) BAG6-KO A549 cells were transfected BAG6-HA expression plasmid or HA-vector. At 24 h post-transfection, the cells were infected with IAV H1N1 (MOI = 1.0). The NP expression were measured by western blotting at different time points post-infection, as indicated (E, upper panels), and densitometry analysis and quantification were performed (E, lower panels). Viral titers in the supernatants were determined by $TCID_{50}$ assay at 6, 12, 24 and 36 h post-infection (F). BAG6-WT A549 cells were infected with IAV H1N1 (MOI = 1.0) as a control. Data presented as means ± SD and are representative of three independent experiments. *$p < 0.05$, **$p < 0.01$, ***$p < 0.001$, Unpaired Student's t test.

interferon-mediated antiviral response. As shown in Fig 4E and 4F, the overexpression of BAG6-HA could also significantly inhibit PR8 replication in Vero cells. These findings indicate that BAG6 is able to inhibit IAV replication independent of IFN-I-mediated innate immune response, and we thus speculate that the mild regulation of IFN-I immune activation presented in BAG6 overexpressing and knockout cells (Fig 4A and 4B) may be due to an indirect effect of BAG6 on virus replication, as numerous studies have shown that IAVs could escape innate antiviral response by inhibiting IFN-I activation.

## BAG6 inhibits the polymerase activity of IAV

To explore the mechanism by which BAG6 inhibits IAV replication, we assessed the impact of BAG6 on the polymerase activity of influenza viruses using a well-established dual-luciferase reporter assay [32,33]. The plasmids for expression of PB2, PB1, PA, and NP genes from various IAV subtype strains, including PR8 (H1N1), CN0606 (H7N9), and SD196 (H9N2), were co-transfected into HEK293T cells along with a pPoll-Luc reporter, respectively. The firefly luciferase levels in the cell lysate reflected the overall transcription and replication activities of viral polymerase complex (RdRp). The results showed that the overexpression of BAG6 inhibited the polymerase activities of H1N1, H7N9, and H9N2 in a dose-dependent manner (Fig 5A). In contrast, the enhanced polymerase activities of all IAV subtypes were observed in BAG6-KO A549 cells compared to the WT control (Fig 5B). The influenza virus polymerase complex is a key enzyme responsible for the replication and transcription of the influenza virus genome [34]. In view of this, we performed qRT-PCR analysis of viral mRNA and vRNA in H1N1-infected cells. The results showed that the mRNA and vRNA levels of both NP and M1 genes were significantly suppressed in BAG6-overexpressing cells (Fig 5C and 5D), and were elevated in BAG6-KO cells (Fig 5E and 5F). These data demonstrate that the BAG6 can inhibit the polymerase activity of various IAV subtypes.

Interestingly, when we examined the control expression of transfected plasmids (PB2, PB1, PA, NP and BAG6) in dual-luciferase reporter assay (Fig 5A and 5B), a remarkably decreasing expression of PB2 protein, but not other viral proteins, from different IAV subtypes were found with the increase of BAG6-HA (Fig 5A right panels). Correspondingly, the expression levels of PB2 also significantly increased in BAG6-KO cells compared to those in BAG6-WT cells (Fig 5B right panels). These results indicate that BAG6 may inhibits the polymerase activity of IAV through regulating PB2 expression.

## BAG6 interacts with viral polymerase subunit PB2 and prevents the assembly of the RdRp complex

To investigate the molecular mechanism by which BAG6 inhibits the polymerase activity of IAV, we examined the potential interaction between BAG6 and viral polymerase subunit proteins, including PB1, PB2, PA and NP. The A549 cells were transfected with BAG6-HA plasmid and then infected with H1N1 (PR8). At 24 h post-infection, the cells were harvested for immunoprecipitation assay with anti-HA agarose. Interestingly, we found that BAG6 was able

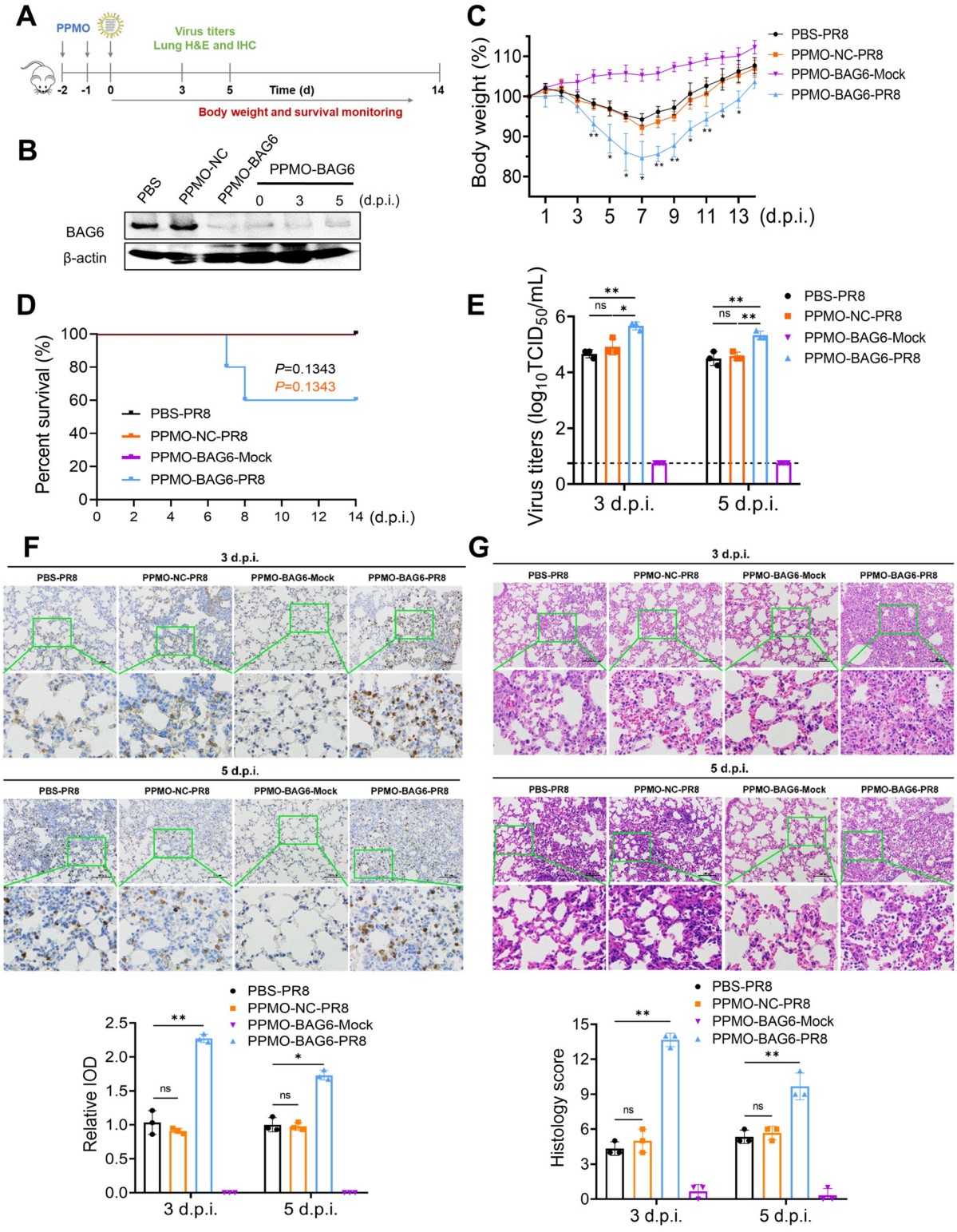

**Fig 3. BAG6 deficiency promotes IAV replication *in vivo*.** (A) Schematic representation of mouse experiments. Six-week-old female BALB/c mice were administered PBS (PBS-PR8), PPMO-non-specific control (PPMO-NC-PR8), or BAG6 targeting PPMO (PPMO-BAG6, 2 groups) intranasally for 2 consecutive days. One group of mice administered PPMO-BAG6 were mock-infected as negative control (PPMO-BAG6-Mock), and the other group of mice administered PPMO-BAG6 were infected with PR8 virus as PPMO-BAG6-PR8. At day 0, two mouse per group were euthanized before infection and the BAG6 expression in lungs were determined. Eleven mouse per group were

infected with PR8 virus (100 $TCID_{50}$) intranasally at day 0. Three mouse per group were euthanized at 3- and 5-days post-infection (d.p.i.). Lungs were collected for viral titer detection and histopathology and immunohistochemistry analysis. Five mouse per group were monitored for survival until 14 days post-infection. All the images and icons used are from open resources. (B) The expression of BAG6 in the indicated lung tissues were detected by western blotting. (C) The body weights of the infected mice were monitored daily for 14 dpi. The mean ± SD of the percent of the initial body weight for each group of mice is shown. (D) Survival of the infected mice was calculated, including the mice that were humanely sacrificed after losing more than 25% of body weight post-infection. (E) Viral titers in the lungs of mice (3 per group) at 3 and 5 dpi. Virus titers were determined by $TCID_{50}$ assays. Each bar is the mean ± SD of the virus titers for each group of mice tissue. (F and G) Mouse lungs were isolated on day 3, and day 5 post infection. Immunohistochemistry (IHC) staining (F, up panels) and H&E staining (G, up panels) of the lung tissues of PBS-PR8 or PPMO-NC-PR8 or PPMO-BAG6-Mock or PPMO-BAG6-PR8 mice at 3- and 5-days post-infection. Scale bars, 100 μm. Relative integral optical density (IOD) and histological score is shown as mean ± SD (n = 3 mice) (down panels). $^*p < 0.05, ^{**}p < 0.01, ^{***}p < 0.001$, Unpaired Student's t test.

to co-immunoprecipitated with all the components of viral ribonucleoprotein (vRNP) complex (PB1, PB2, PA and NP), but not with other viral proteins, such as M1 (Fig 6A). We next wished to verify the specific viral protein that directly interact with BAG6. To avoid that viral polymerase subunit proteins may be immunoprecipitated by BAG6-HA through indirect binding of vRNP complex, we performed immunoprecipitation assay in A549 cells co-transfected with BAG6-HA and PB1-Flag, or PB2-Flag, or PA-Flag, or NP-Flag, respectively. The M1-Flag was used as a negative control. The results showed that BAG6-HA was only able to interact with PB2-Flag (Fig 6B, lane 3), but not with PB1-Flag, PA-Flag, or NP-Flag individually (Fig 6B, lanes 4–6). Meanwhile, BAG6-HA was also readily pulled down by PB2-Flag-agarose in the co-IP assay (Fig 6C), confirming the interaction between BAG6 and PB2. Subsequently, we further verified the association of endogenous BAG6 and PB2 using anti-BAG6 mAb (Fig 6D). The immunofluorescence staining showed that while the PB2 alone mainly localized in the nuclei, BAG6 could co-localize with PB2 in both nuclei and cytoplasm of cells (Fig 6E).

PB2 protein is an essential component of the influenza virus polymerase. It interacts with PB1 through its short N-terminal peptide fragment but does not directly interact with PA [35–37], forming a compact, spherical structure of RNA polymerase [8,38,39]. We next investigated whether the interaction between BAG6 and PB2 would affect the binding of PB2 and PB1. The Co-IP assay revealed that BAG6 protein could bind to the N-terminus of PB2 protein (amino acids 1–247) (Fig 6F, lane 3). Whereas, a reduced binding level of PB1 in PB2-Flag-agarose was observed in BAG6-HA overexpressing cells compared to that in the control cells (Fig 6G, compared lane 5 and 6) and the immunoprecipitated PB1 decreased in a BAG6-dose-dependent manner (Fig 6H), suggesting that BAG6 could compete with PB1 to bind PB2, thereby inhibiting the formation of viral polymerase complex.

## BAG6 induces PB2 degradation through K48-linked ubiquitination pathway

BAG6 is reported as a proteasomal substrate associated protein and participates in the efficient proteasome targeting and ubiquitin degradation primarily mediated by E3 ligase RNF126 [40–42]. We then examined the stability and degradation of PB2 protein in BAG6-overexpressing cells. A cycloheximide chase assay showed that ectopic expression of BAG6-HA accelerated the degradation of Flag-PB2 protein in A549 cells in comparison with the vector control (Fig 7A). The reduced PB2 protein levels by BAG6-HA in both Flag-PB2-transfected cells (Fig 7B) and PR8-infected cells (Fig 7C) were restored to the control level by the treatment of proteasome inhibitor MG132, but not by lysosome inhibitor Chloroquine (CQ), indicating that BAG6 likely induces proteasome degradation of PB2. We next explored the effect of BAG6 on

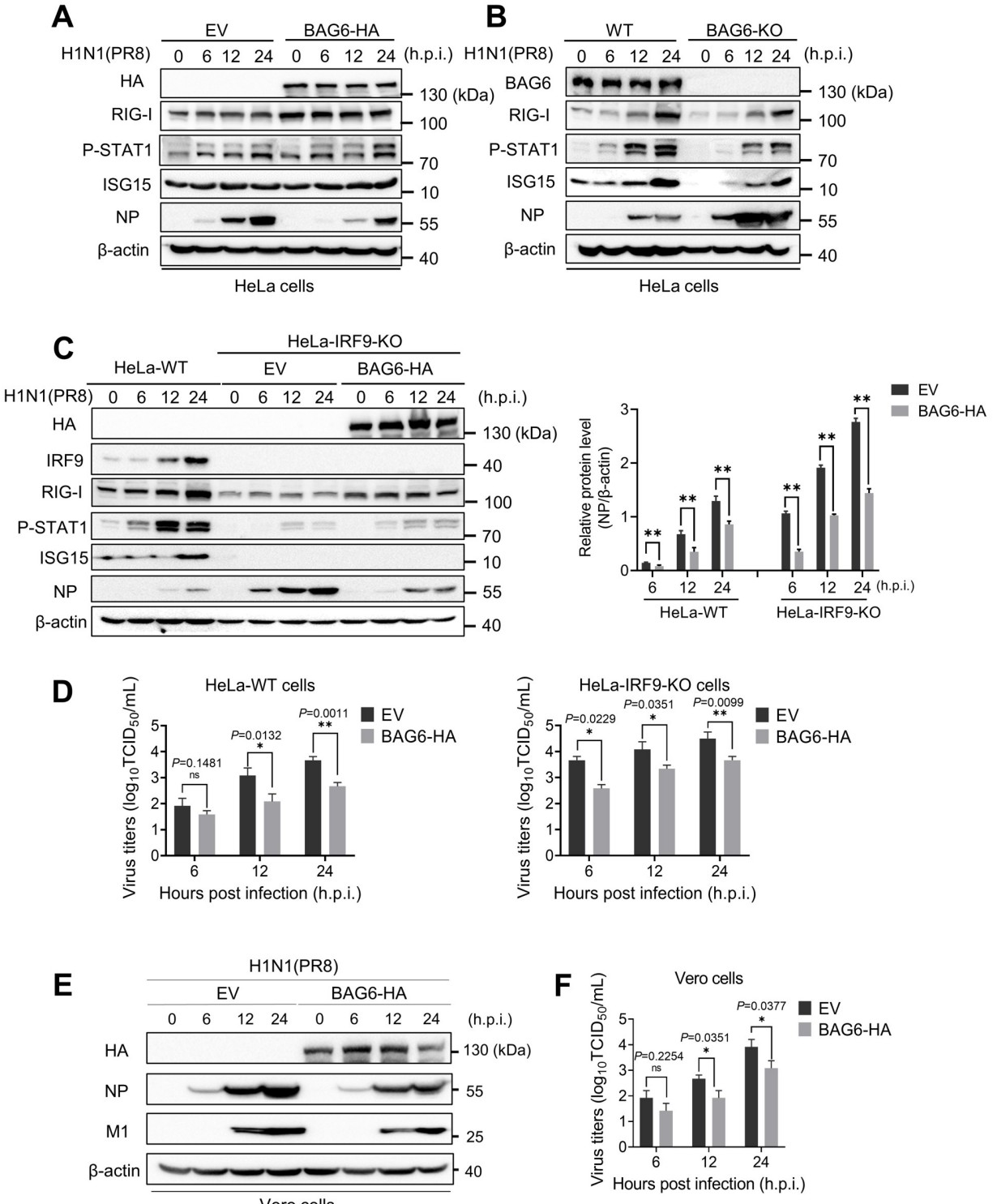

**Fig 4. BAG6 inhibits IAV replication independent of IFN-mediated innate immune pathways.** (A) HeLa cells were transfected with BAG6-HA or Empty vector (EV) plasmids. At 24 h post-transfection, the cells were infected with IAV H1N1 (MOI = 1.0). The viral NP and innate immunity-associated protein expression were measured by western blotting at different time points postinfection, as indicated. (B) BAG6-KO and BAG6-WT HeLa cells were infected with IAV H1N1 (MOI = 1.0), and the NP and innate immunity-associated protein expression were determined by western blotting at the indicated time points. (C) IRF9-KO Hela cells were transfected BAG6-HA expression plasmid or HA-vector. At 24 h post-transfection, the cells were infected with IAV H1N1 (MOI = 1.0). The NP and innate immunity-associated protein expression were measured by western blotting at different time points post-infection, as indicated (C, left panels), and densitometry

analysis and quantification were performed (C, right panels). (D) HeLa-WT (left panels) or HeLa-IRF9-KO (right panels) cells were transfected with BAG6-HA or Empty vector (EV) plasmids. At 24 h post-transfection, the cells were infected with IAV H1N1 (MOI = 1.0). Viral titers in the supernatants were determined by TCID$_{50}$ assay at 6, 12 and 24 h post-infection. (E and F) Vero cells were transfected with BAG6-HA or Empty vector (EV) plasmids. At 24 h post-transfection, the cells were infected with IAV H1N1 (MOI = 1.0). The viral NP and M1 expression were measured by western blotting at different time points postinfection, as indicated (E). Viral titers in the supernatants were determined by TCID$_{50}$ assay at 6, 12 and 24 h post-infection (F). Data presented as means ± SD and are representative of three independent experiments. *$p < 0.05$, **$p < 0.01$, ***$p < 0.001$, Unpaired Student's t test.

PB2 ubiquitination by a ubiquitination assay, and found that the polyubiquitination level of PB2 was up-regulated by BAG6 overexpression (Fig 7D).

The type of ubiquitin chain linked to a protein determines its fate. K48- and K63-linked polyubiquitin chains are the two most abundant polyubiquitin chain types. K48 chains tag substrates for proteasomal degradation, whereas K63 chains regulate the functions of target proteins [43]. We further characterized which type of ubiquitin chains of PB2 was regulated by BAG6 in A549 cells co-transfected with BAG6-HA and Flag-PB2, as well as K48 or K63 ubiquitin plasmids, respectively. As shown in Fig 7E, BAG6 promoted the K48-linked ubiquitination of PB2 but did not influence the K63-linked ubiquitination of PB2 (Fig 7E), indicating that BAG6 induced the K48-linked ubiquitination and degradation of PB2.

Ubiquitin typically couples to the internal lysine residues on substrate proteins to form an isopeptide bond. Since eighteen lysine residues have been reported to be conserved among IAV PB2 proteins [44], we next identify the specific lysine residues on PB2 ubiquitinated by BAG6 using site-directed mutagenesis. We generated PB2 mutants containing individual lysine to arginine mutations, and assessed the ubiquitination of each lysine-mutated PB2 proteins in cells transfected with HA-BAG6. As shown in Fig 7F, mutation of K189R strongly reduced PB2 ubiquitination upon expression of HA-BAG6 (Fig 7F, lane 11), suggesting that residue K189 is the main target for ubiquitination by BAG6. Consistently, the mutation of K189R in PB2 prevented it from being degraded by HA-BAG6 in comparison with the PB2-WT protein (Fig 7G). Since K189 is highly conserved in PB2 of different IAV subtypes (S1 Table), these data underscore that BAG6-mediated polyubiquitination and degradation of PB2 is a common host defense mechanism against IAV infection.

## The N-terminus of BAG6 interacts with PB2 and inhibits IAV replication

BAG6 is a long multi-domain protein, consisting of an amino-terminal ubiquitin-like (UBL) domain, a central proline-rich segment, a zinc finger-like domain and a carboxyl-terminal BAG domain (Fig 8A). To identify the functional domain of BAG6 that is crucial for interacting with PB2 and restricting IAV infection, we constructed a series of truncated BAG6 mutants (as indicated in Fig 8A) and measured the interaction between PB2 and these mutants in HEK293T cells. The Co-IP assay showed that the mutants 1-255aa and 1-753aa at the N-terminus of BAG6 retained the ability to interact with PB2 to the level of full-length BAG6 (Fig 8B and 8C, lanes 4 and 5), whereas other truncated mutants lacking the N-terminus lost their ability to bind PB2 (Fig 8B and 8C, lanes 6–8). More interestingly, the immunofluorescence staining showed that the PB2-Flag alone mainly localized in the nuclei, but the N-terminal truncation of BAG6 (BAG6 1:255-HA), which lacks a nuclear localization signal (NLS), could co-localize with PB2 and induce its accumulation in the cytoplasm (Fig 8D). These results demonstrated that the 1-255aa at the N-terminus of BAG6 is necessary and sufficient for interacting with PB2 and might be responsible for targeting it for cytoplasm accumulation. We also examined the effect of truncation mutants (BAG6 1:255-HA or 641:1132aa-HA) on IAV replication and viral polymerase activity by western blotting, TCID$_{50}$ assay, and pPoll-Luc reporter

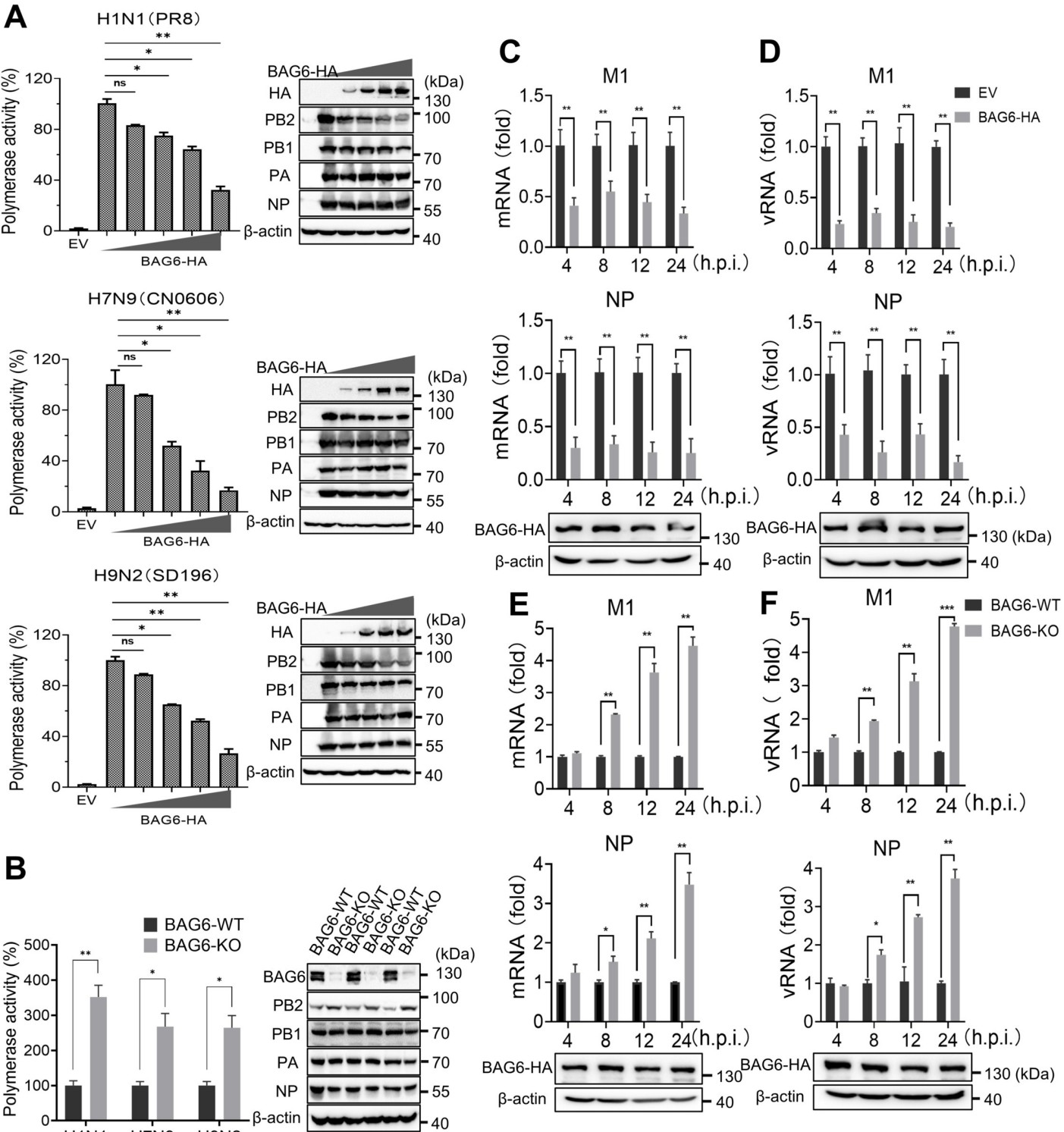

**Fig 5. BAG6 inhibits the polymerase activity of IAV.** (A) HEK293T cells were co-transfected with pPolI-Luc reporter plasmid and expression plasmids containing viral PB1, PB2, PA, and NP of H1N1, H7N9 or H9N2 along with a graded dose of BAG6 or empty vector. Renilla luciferase was used as an internal control. Luciferase activity of the viral polymerase was determined at 24 h post-transfection (left panels). Western blottings show the expression of all transfected proteins (right panels) (B) BAG6-KO or BAG6-WT A549 cells were co-transfected with pPolI-Luc reporter plasmid and expression plasmids containing viral PB1, PB2, PA, and NP of H1N1/H7N9/H9N2. Renilla luciferase was used as an internal control. Luciferase activity of the viral polymerase was determined at 24 h post-transfection (left panel). Western blottings show the expression of all transfected proteins (right panel). (C and D) A549 cells was transfected with BAG6 expression plasmid or empty vector.

At 24 h post-transfection, the cells were infected with IAV H1N1 (MOI = 1.0). At 4, 8, 12 and 24 h post-infection, the mRNA (C) and vRNA (D) level of M1 and NP were measured by quantitative real-time PCR, respectively. The expression of BAG6-HA was monitored by western blotting. (E and F) BAG6-KO or BAG6-WT A549 cells were infected with IAV H1N1 (MOI = 1.0). At 4, 8, 12 and 24 h post-infection, the mRNA (E) and vRNA (F) level of NP and M1 were measured by quantitative real-time PCR, respectively. The expression of BAG6-HA was monitored by western blotting. Data presented as means ± SD and are representative of three independent experiments. *$p < 0.05$, **$p < 0.01$, ***$p < 0.001$, Unpaired Student's t test.

assay, respectively. The result showed that the overexpression of BAG6 1:255-HA, but not BAG6 641:1132-HA, significantly suppressed the expression of PB2 and NP (Fig 8E), reduced viral titers (Fig 8F), and inhibited viral polymerase activity (Fig 8G) in H1N1-infected cells. Meanwhile, the viral protein expression (Fig 8H) and virus titers (Fig 8I) up-regulated in BAG6-KO cells were also restored to the control levels (BAG6-WT) by BAG6 1:255-HA over-expression. These results demonstrate that BAG6 blocks IAV polymerase activity and viral replication via its N-terminal 1-255aa region.

## The synergistic effect of UBL domain (17-92aa) and PB2-binding domain (124-186aa) of BAG6 inhibits polymerase activity and restricts IAV replication

To further determine the PB2-binding sites of BAG6, we analyzed the protein structure of BAG6 from AlphaFold2 database and predicted the corresponding interaction domain of BAG6 and PB2 using an online software ColabFold. As shown in Fig 9A, BAG6 associates with the N-terminus of PB2 through its 124–186 amino acids, which at least partially occupies the spatial position of PB1-PB2 interaction, whereas the Ub-like domain (17-92aa) at the N-terminus of BAG6 does not exhibit a direct interaction with PB2 in structure. According to the previous reports [40,41,43,45], the UBL domain is mainly responsible for recruiting ubiquitination machinery for efficient ubiquitination and degradation of various protein substrates. We, therefore, hypothesize that BAG6 may function as a transient platform that recruits the ubiquitin ligase by UBL domain and binds PB2 substrate by 124-186aa region during IAV infection, promoting PB2 degradation and IAV restriction. To verify this hypothesis, we constructed the corresponding truncation mutant plasmids, including BAG6 1:92-HA and 92:255-HA, as well as a deletion mutant lacking 124-186aa (BAG6 del 124:186-HA), as indicated in Fig 9B. The Co-IP assay showed that only BAG6 92:255-HA was readily immunoprecipitated with PB2-Flag, but other mutants lacking 124-186aa region (1:92-HA and del 124:186-HA) could not bind to PB2 (Fig 9C), confirming the residues 124–186 as PB2-binding region of BAG6. Interestingly, compared with BAG6 1:255-HA, the ectopic expression of BAG6 92:255-HA had an obvious but reduced inhibitory effect on viral NP protein expression (Fig 9D), viral titers (Fig 9E), and viral polymerase activity (Fig 9F) in H1N1-infected cells, whereas BAG6 1:92-HA and del 124:186-HA completely failed to inhibit IAV replication and polymerase activity. These findings indicate that the PB2-binding region of BAG6 is necessary but insufficient for efficient IAV restriction. It may exert a reduced antiviral effects through binding PB2 and disrupting RdRp complex assembly, but fail to recruit ubiquitination machinery for PB2 degradation. These data demonstrate that both the UBL domain (17-92aa) and PB2-binding region (124-186aa) are required for highly effective PB2 degradation and IAV restriction.

## Discussion

IAVs are among the most common contagious pathogens to cause severe respiratory infections. Host cells have developed multiple strategies to defend against IAV infection. This study

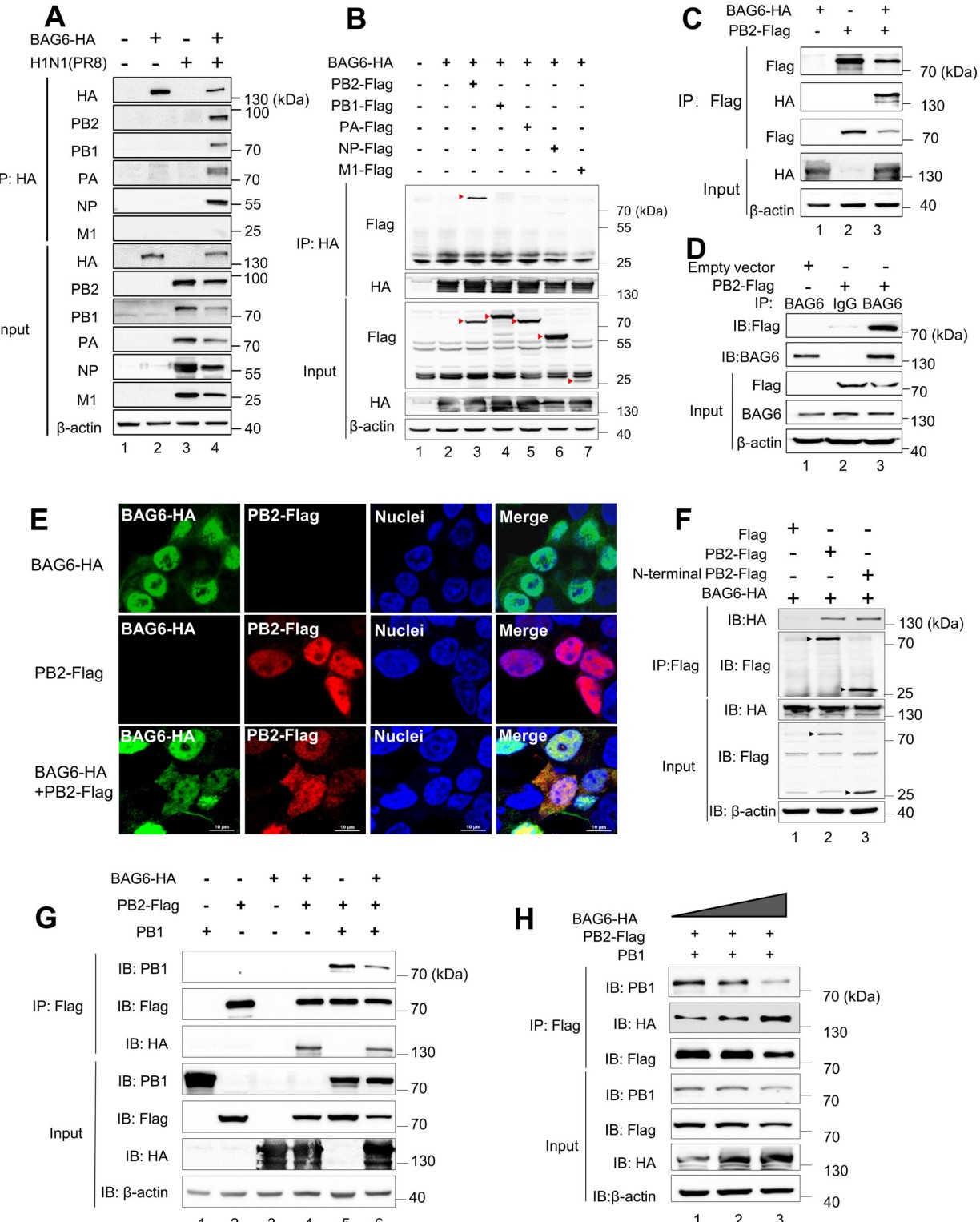

**Fig 6. BAG6 interacts with viral polymerase subunit PB2 and prevents the RdRp complex assembly.** (A) A549 cells were transfected with/without BAG6-HA expression plasmid and then were infected with IAV H1N1 virus (MOI = 1.0). Cell lysates were immunoprecipitated with an anti-HA mAb and then analyzed by western blotting. (B) HEK293T cells were transfected with BAG6-HA individually or in combination with plasmids that expressed PB2-Flag, PB1-Flag, PA-Flag, NP-Flag or M1-Flag. Cell lysates were immunoprecipitated with anti-HA mAb and then analyzed by western blotting. (C) HEK293T cells were co-transfected with BAG6-HA and PB2-Flag plasmids for 24 h. Cell lysates were

immunoprecipitated with anti-Flag mAb and then analyzed by western blotting. (D) HEK293T cells were transfected with PB2-Flag or empty vector for 24 h. Cell lysates were immunoprecipitated with anti-BAG6 mAb or IgG, and then analyzed by western blotting. (E) A549 cells were transfected with BAG6-HA and/or PB2-Flag. After 24 h post transfection, BAG6-HA (green) and PB2-Flag (red) were visualized by immunofluorescence with HA or Flag mAb. The cell nuclei were stained with DAPI (blue). (F) HEK293T cells were co-transfected with BAG6-HA and PB2-Flag or N-terminal PB2-Flag expression plasmids (residues 1–247). After 24 h post transfection, cell lysates were immunoprecipitated with anti-Flag mAb and then analyzed by western blotting. (G) HEK293T cells were transfected with BAG6-HA, in combination with/without PB2-Flag and PB1-Flag expression plasmids. After 24 h post transfection, cell lysates were immunoprecipitated with anti-Flag mAb and then analyzed by western blotting. (H) HEK293T cells were transfected with PB2-Flag, PB1, and a gradient dose of BAG6-HA expression plasmids. After 24 h post transfection, cell lysates were immunoprecipitated with anti-Flag mAb and then analyzed by western blotting.

identified BAG6 as a novel intrinsic antiviral factor to negatively regulate the replication of multiple IAV subtypes, including human H1N1, H5N1 and avian H7N9, H9N2. BAG6 specifically interacts with the N-terminus of viral polymerase subunit protein PB2, competitively interrupts the assembly of functional RdRp complex and induces the ubiquitination degradation of PB2, which attenuate viral polymerase activity and IAV replication in mammalian cells and pathogenicity in mouse model (Fig 10). These findings reveal a new antagonism mechanism between pathogens and hosts, and highlight a potent antiviral activity of BAG6 by targeting the key viral protein PB2.

PB2 protein of IAV is critical for RdRp complex to catalyze the viral transcription and replication, and acts as a key determinant for mammalian adaptation of avian influenza virus. PB2 is responsible for binding the polymerase complex to host cell RNA and initiating viral transcription. It contains a conserved N-terminal fragment that interacts with PB1 to form the polymerase complex, and a central cap-binding domain that recognizes the 5' cap structure of host cell mRNA, allowing the polymerase to "steal" the cap and use it to prime viral transcription [34]. Besides, the C-terminal part of PB2 contains the so-called 627 domain (residues 535–684), which participates in host adaptation and virulence by influencing the binding of NLS to importins [46]. Multiple host proteins have been found to restrict IAV replication by targeting PB2. For instance, TRIM35 mediates the protection against influenza infection by activating TRAF3 and inducing K48-linked ubiquitination and degradation of PB2 [44]. PIAS restricts IAV replication and virulence through mediating PB2 SUMOylation and reducing its stability [47]. In addition, ANP32A inhibits IAV replication in mammalian cells through differential regulation of the activity of viral polymerases carrying PB2-627K (human) or PB2-627E (avian) signature [12,48]. Here, we found that BAG6 associated with PB2, thereby perturbing its interaction with PB1 and inducing its ubiquitination degradation. Notably, BAG6 recognizes the N-terminus of PB2 and promotes PB2 ubiquitination at its K189 residue, which is highly conserved among avian and human influenza strains from Global Initiative on Sharing Avian Influenza Data (GISAID, www.epicov.org) (S1 Table), providing the mechanistic basis that BAG6 acts as a potent antiviral factor during both human- and avian-origin IAV infection.

BAG6 is a multifunctional protein in host cell and participates in a variety of physiological and pathological processes [19]. BAG6 functions as an essential factor in ubiquitin-mediated metabolism of neo-synthesized defective polypeptides and misfolded/mislocalized proteins [49–51]. It can capture these defective or mislocalized substrates by interacting with their unprocessed or non-inserted hydrophobic domains [45], and at the same time, recruit the E3 ligases, primarily RNF126, to its N-terminal UBL domain for efficient ubiquitination and degradation [40]. Besides, BAG6 is necessary for DNA damage-induced apoptosis by controlling the p53 signaling pathway [52]. It promotes p53 transactivation by stabilizing its interaction with the acetyltransferase p300 and enhancing p53 acetylation during apoptosis and autophagy [21,22,53]. A recent study reported that BAG6 was capable of regulating the activation of

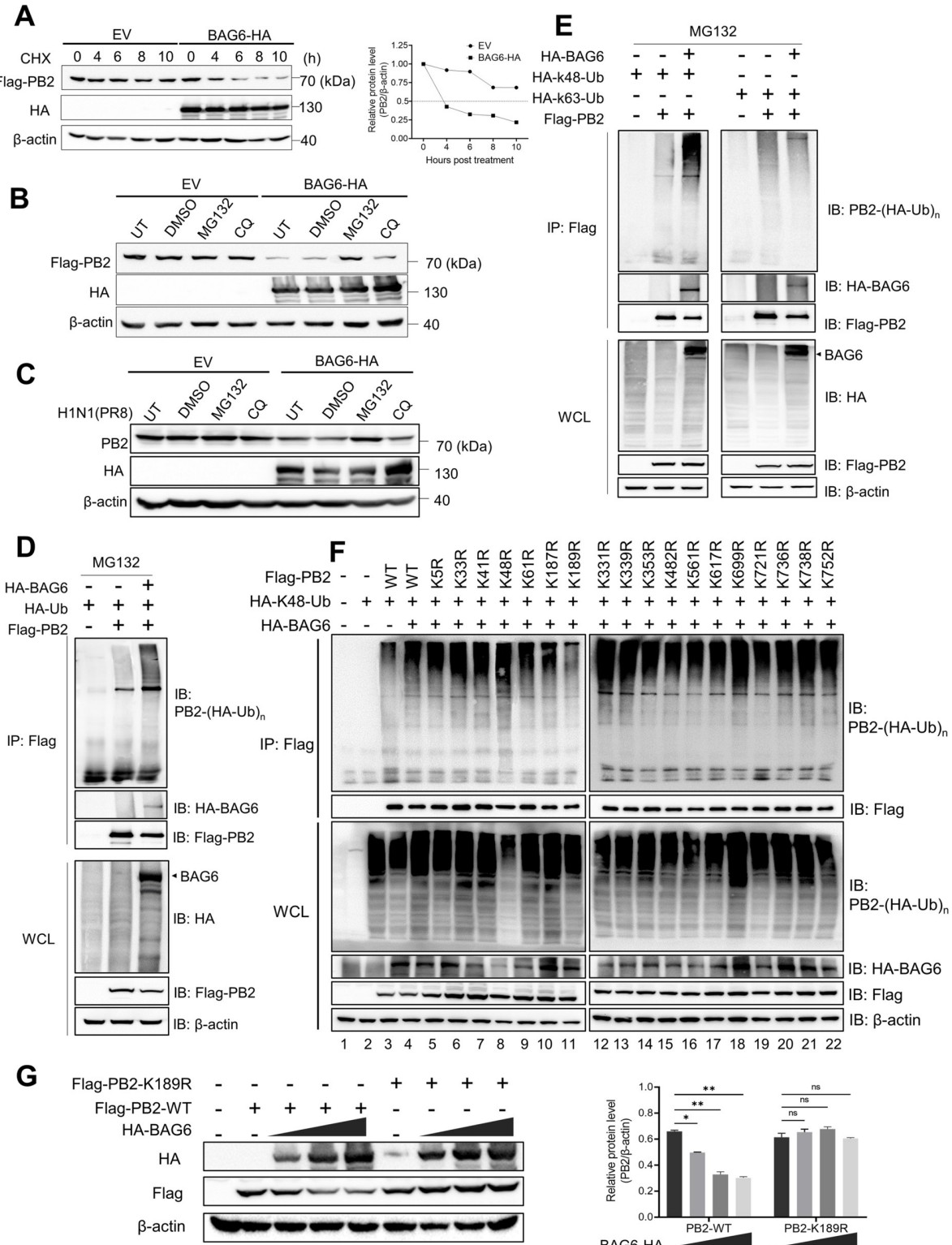

**Fig 7. BAG6 induces viral PB2 degradation through K48-linked ubiquitination pathway.** (A) HEK293T cells transfected with PB2-Flag expression plasmid, in combination with empty vector or BAG6-HA plasmid were treated with 50 μg/mL CHX for the indicated time periods. The protein levels of PB2 were determined by western blotting (left panel) and quantified using grayscale analysis (right panel). (B) HEK293T cells transfected with PB2-Flag expression plasmid, in combination with empty vector or BAG6-HA plasmid were treated with DMSO, MG132 or CQ for 8 h. The protein levels of PB2-Flag were determined by western blotting. (C) A549 cells were transfected with

empty vector or BAG6-HA plasmid, respectively, and then infected with PR8 virus (MOI = 1) for 16 h post-infection, and then treated with DMSO, MG132 or CQ for 8 h, respectively. The protein levels of PB2 were determined by western blotting. (D) HEK293T cells transfected with HA-BAG6, Flag-PB2 and HA-Ub plasmids were immunoprecipitated with M2/Flag antibody, followed by western blot analysis using anti-HA monoclonal antibody. All cells were treated with 20 μM MG132 for 4 h before being harvested. (E) HEK293T cells transfected with HA-BAG6, Flag-PB2 and HA-K48-Ub (left panel) or HA-K63-Ub (right panel) plasmids were immunoprecipitated with M2/Flag antibody, followed by western blot analysis using anti-HA monoclonal antibody. All cells were treated with 20 μM MG132 for 4 h before being harvested. (F) HEK293T cells transfected with HA-BAG6, HA-K48-Ub and Flag-PB2 or its mutants were immunoprecipitated with M2/Flag antibody, followed by western blot analysis using anti-HA monoclonal antibody. All cells were treated with 20 μM MG132 for 4 h before being harvested. (G) HEK293T cells transfected with Flag-PB2 or Flag-PB2-K189R plasmid, in combination with increasing amounts of BAG6-HA. The protein levels of Flag-PB2 and HA-BAG6 were determined by western blotting (left panel) and the relative protein levels of PB2 were quantified by densitometry and normalized to the levels of β-actin (right panel).

IFN-I by inhibiting RIG-I/VISA-mediated recognition of RNA ligand [29]. However, the role of BAG6 in virus infection remains opaque. In this study, we provide evidence that BAG6 can potently inhibit influenza virus replication *in vitro* and *in vivo*. Further investigation found that the overexpression of BAG6 neither affected cell apoptosis during IAV infection (S4 Fig), nor relied on the IFN-I antiviral response to inhibit IAV replication (Fig 4), whereas BAG6 directly interacts with viral polymerase subunit PB2 and targets it for ubiquitination degradation. These findings reveal a novel mechanism by which BAG6 serves as an antiviral factor to specifically target viral protein through its protein quality control function.

In the recent years, there has been increased interest in harnessing targeted protein degradation (TPD) technology in the development of antiviral therapies by inducing the degradation of either viral or host-related protein targets, but the off-target side effects and toxicity greatly limit its use [54–57]. In this study, we found the N-terminus of host protein BAG6 contains a ubiquitin-like domain (UBL, 17-92aa), which was previously reported to recruit E3 ligase RNF126 and ubiquitinates BAG6-associated clients [40], and a PB2-binding domain (124-186aa), which was proved to interact with viral polymerase subunit PB2 protein during IAV infection. Interestingly, although the BAG6:92-255aa mutant lacking the UBL domain weakened BAG6-meditated antiviral activity against IAV, it still significantly restricted viral polymerase activity and replication (Fig 9E and 9F). This mirrors the findings that BAG6 could suppress the viral polymerase activity by competing with PB1 for PB2 binding and interfering with RdRp assembly (Fig 6E and 6F), even in the situation with PB2 degradation resistance. These data indicate that both UBL and PB2-binding domains are essential for the efficient PB2 degradation and BAG6-mediated antiviral activity. This unique structure thus allows BAG6 naturally possessing a proteolysis-targeting function. By targeting viral PB2, BAG6 is more likely to become a potential candidate for antiviral drug development.

## Materials and methods

### Ethics statement

The laboratory animal usage license number is SYXK-Beijing-2018-0038, certified by Beijing Association for Science and Technology. All animal research was performed in compliance with the Beijing Laboratory Animal Welfare and Ethics guidelines, as issued by the Beijing Administration Committee of Laboratory Animals, and the China Agricultural University (CAU) Institutional Animal Care and Use Committee guidelines (ID: SKLAB-B-2010-003) approved by the Animal Welfare Committee of CAU. All experiments with live viruses were performed in Biosafety Level II laboratory (BSL-2) in CAU. All animal research involved in this study was approved by the Animal Welfare Committee of CAU (No.AW03503202-2-2).

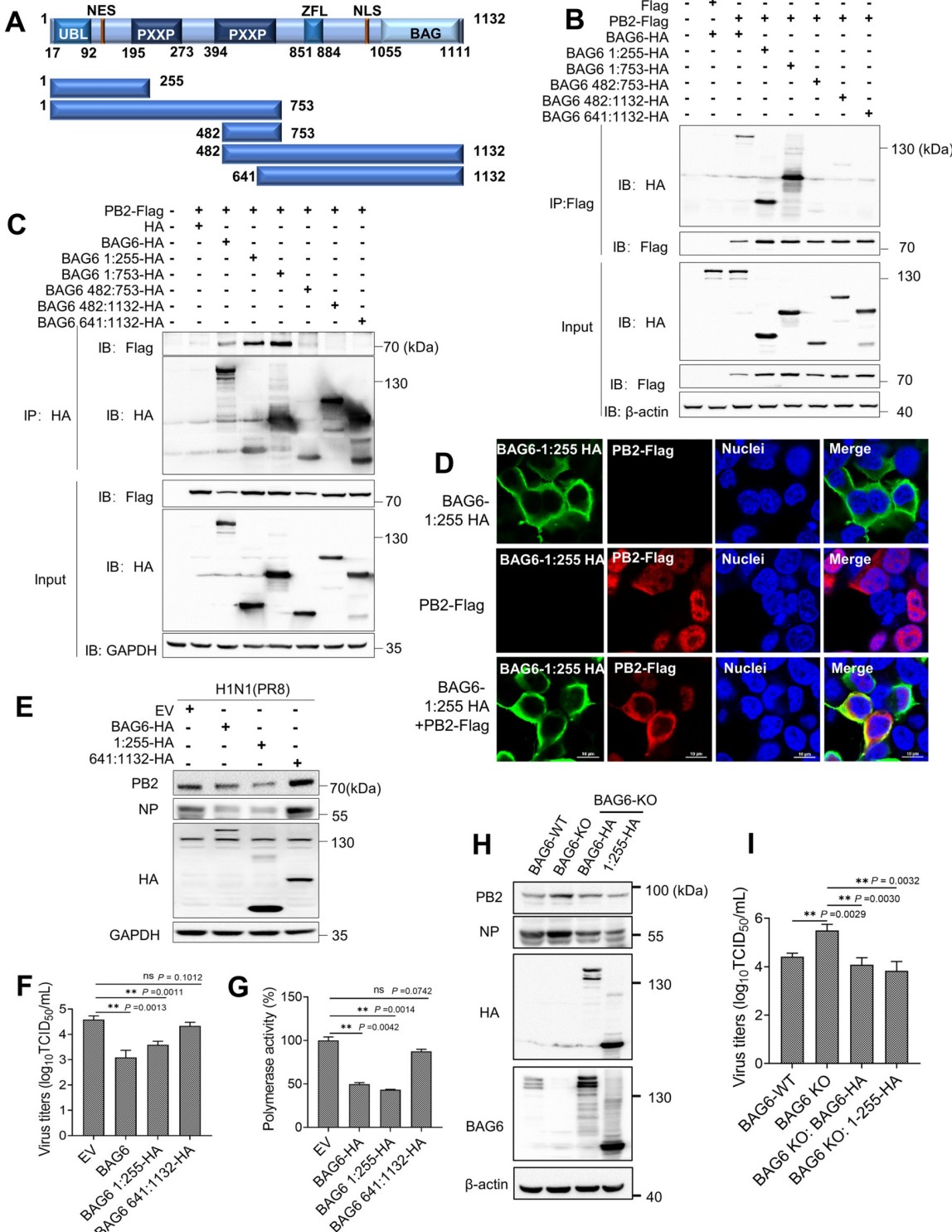

**Fig 8. 1–255aa at the N-terminus of BAG6 interacts with PB2 and potently inhibits IAV replication.** (A) Schematic diagram illustrating the full-length and truncated mutants of BAG6. UBL: ubiquitin-like domain; PXXP: proline-rich region; NES: nuclear export signal; NLS: nuclear localization signal; ZFL: zinc-finger like domain; BAG: Bcl-2-associated athanogene. (B and C) HEK293T cells transfected with the indicated plasmids were immunoprecipitated with anti-M2/Flag antibody (B) or HA antibody (C), followed by western blot analysis. (D) A549 cells were transfected with BAG6-HA and/or BAG6 1:255-HA plasmids. After 24 h post transfection, BAG6-HA (green), BAG6 1:255-HA (green) and PB2-Flag (red) were visualized by immunofluorescence with HA or Flag mAb. The cell

nuclei were stained with DAPI (blue). (E and F) A549 cells were transfected with BAG6-HA or BAG6 1:255-HA or BAG6 641:1132-HA or empty vector for 24 h, and then infected with IAV H1N1 (MOI = 1.0). The viral PB2 and NP proteins were measured by western blotting (E), the viral titers in the supernatants were determined by $TCID_{50}$ assay (F). (G) HEK293 cells were co-transfected with pPolI-Luc reporter plasmid and expression plasmids containing viral PB1, PB2, PA, and NP of H1N1 along with BAG6 1:255-HA or BAG6 641:1132-HA. Renilla luciferase was used as an internal control. Luciferase activity of the viral polymerase was determined at 24 h post-transfection. (H and I) BAG6-WT or BAG6-KO A549 cells were transfected with BAG6-HA or BAG6 1:255-HA for 24 h, and then infected with IAV H1N1 (MOI = 1.0). The viral PB2 and NP proteins were measured by western blotting (H), the viral titers in the supernatants were determined by $TCID_{50}$ assay (I). Data presented as means ± SD and are representative of three independent experiments. $*p < 0.05$, $**p < 0.01$, $***p < 0.001$, Unpaired Student's t test.

## Cells and viruses

Human lung adenocarcinoma epithelial cells (A549), HeLa cell, Madin-Darby canine kidney cells (MDCK), Vero cell and human embryonic kidney cells (HEK293T) were purchased from ATCC. IRF9-KO HeLa cells were generated as previous indicated [30,31]. BAG6 KO A549 cells and BAG6 KO HeLa cells were generated using CRISPR/Cas 9-based knockout of BAG6 gene. In brief, the BAG6 gene target sequences, #1 5'- CACCGCGGTACTGGTACTAT-CATT-3' and #2 5'- CACCAGGCTCCTCCACAGCGGTAC-3', were inserted into the guide RNA (gRNA) expression cassette of the pUC19 vector, which also contains an expression cassette of Cas9. The pUC19 plasmid containing the BAG6 target sequence was then transfected into A549 or HeLa cells. The transfected cells were trypsinized 24 h later into single cells, which were diluted and inoculated into a 96-well plate by infinite dilution. Each colony was individually propagated in a 96-well plate, and the knockout of BAG6 expression was confirmed by western blotting. Cells were cultured in DMEM supplemented with 10% v/v heat-inactivated fetal bovine serum (10099141C, Gibco, USA), 100 U mL$^{-1}$ penicillin and 100 µg mL$^{-1}$ streptomycin at 37°C in a 5% $CO_2$ atmosphere.

Influenza A/Puerto Rico/8/1934 (PR8, H1N1, accession HA number: CY045764), A/chicken/China/0606-13/2017 (CN0606, H7N9, accession HA number: MK530478), and A/chicken/Shandong/196/2011 (SD196, H9N2, accession HA number: KR002651) and A/Anhui/1/2005 (AH1, H5N1, accession HA number: HM172104) viruses were maintained in the laboratory. All experiments using non-H5 live viruses were performed in the Biosafety Level II laboratory (BSL-2), and experiments using H5 viruses were performed in the Biosafety Level III Laboratory (BSL-3) in CAU.

## Plasmids construction and transfection

Protein expression plasmids BAG6-HA, BAG6 1:255-HA, BAG6 1:753-HA, BAG6 482:753-HA, BAG6 482:1126-HA, BAG6 642:1126-HA, PB2-Flag, N-terminal PB2-Flag, PB1-Flag, PA-Flag, NP-Flag, M1-Flag, PB2-K5R-Flag, PB2-K33R-Flag, PB2-K41R-Flag, PB2-K48R-Flag, PB2-K61R-Flag, PB2-K187R-Flag, PB2-K189R-Flag, PB2-K331R-Flag, PB2-K339R-Flag, PB2-K353R-Flag, PB2-K482R-Flag, PB2-K561R-Flag, PB2-K617R-Flag, PB2-K699R-Flag, PB2-K721R-Flag, PB2-K736R-Flag, PB2-K738R-Flag, PB2-K752R-Flag, Ub-HA, K48-Ub-HA and K63-Ub-HA were generated by subcloning the corresponding coding segment into the HA-pRK5 or Flag-pRK5 empty vector by homologous recombinase-mediated recombination (S1 Table). Plasmid transfections in HEK293T or A549 cells were performed using jetPRIME (101000046, Polyplus, France).

## Antibodies and reagents

Antibody concentrations for immunoprecipitation and immunoblotting were determined empirically. All immunoblot antibodies were diluted as specified in 5% w/v BSA-TBST.

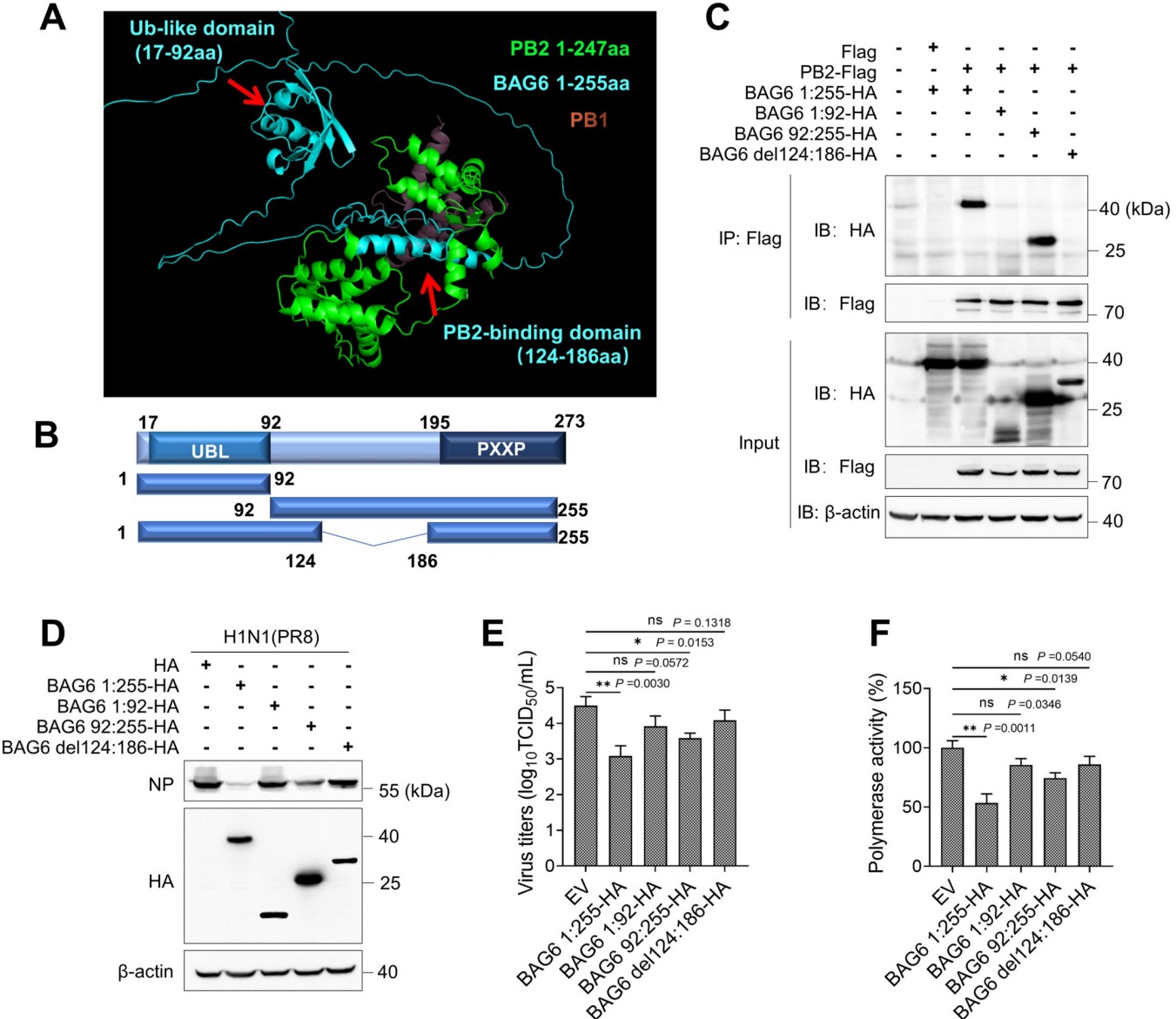

**Fig 9. The UBL domain (17-92aa) and PB2-binding domain (124-186aa) are responsible for effective IAV restriction.** (A) The predicted interaction domain of BAG6 and PB2. The protein structure of PB2 and BAG6 from AlphaFold2 database and the predicted interaction complex from ColabFold online software, showing that BAG6 124–186 amino acids is the key region for binding to the N-terminus of PB2 protein, which competitively prevents PB1-PB2 interaction. The N-terminus of BAG6 (residues 1–255, blue) and the N-terminus of PB2 (residues 1–247, green) from ColabFold online prediction, the interaction region of PB1 in complex with PB2 (residues 678–757, light red) from PDB:3A1G. (B) Schematic diagram illustrating the truncated and deleted mutants of BAG6 based on the predicted interaction domain. UBL: ubiquitin-like domain; PXXP: proline-rich region. (C) HEK293T cells transfected with the indicated plasmids were immunoprecipitated with M2/Flag antibody, followed by western blot analysis. (D and E) A549 cells were transfected with BAG6 1:255-HA or BAG6 1:92-HA or BAG6 92:255-HA or BAG6 del124:186-HA empty vector for 24 h, and then infected with IAV H1N1 (MOI = 1.0). The viral NP proteins were measured by western blotting (D), the viral titers in the supernatants were determined by $TCID_{50}$ assay (E). (F) HEK293T cells were co-transfected with pPolI-Luc reporter plasmid and expression plasmids containing viral PB1, PB2, PA, and NP of H1N1 along with BAG6 1:255-HA or different truncated mutants of BAG6 or empty vector. Renilla luciferase was used as an internal control. Luciferase activity of the viral polymerase was determined at 24 h post-transfection. Data presented as means ± SD and are representative of three independent experiments. $^*p < 0.05$, $^{**}p < 0.01$, $^{***}p < 0.001$, Unpaired Student's t test.

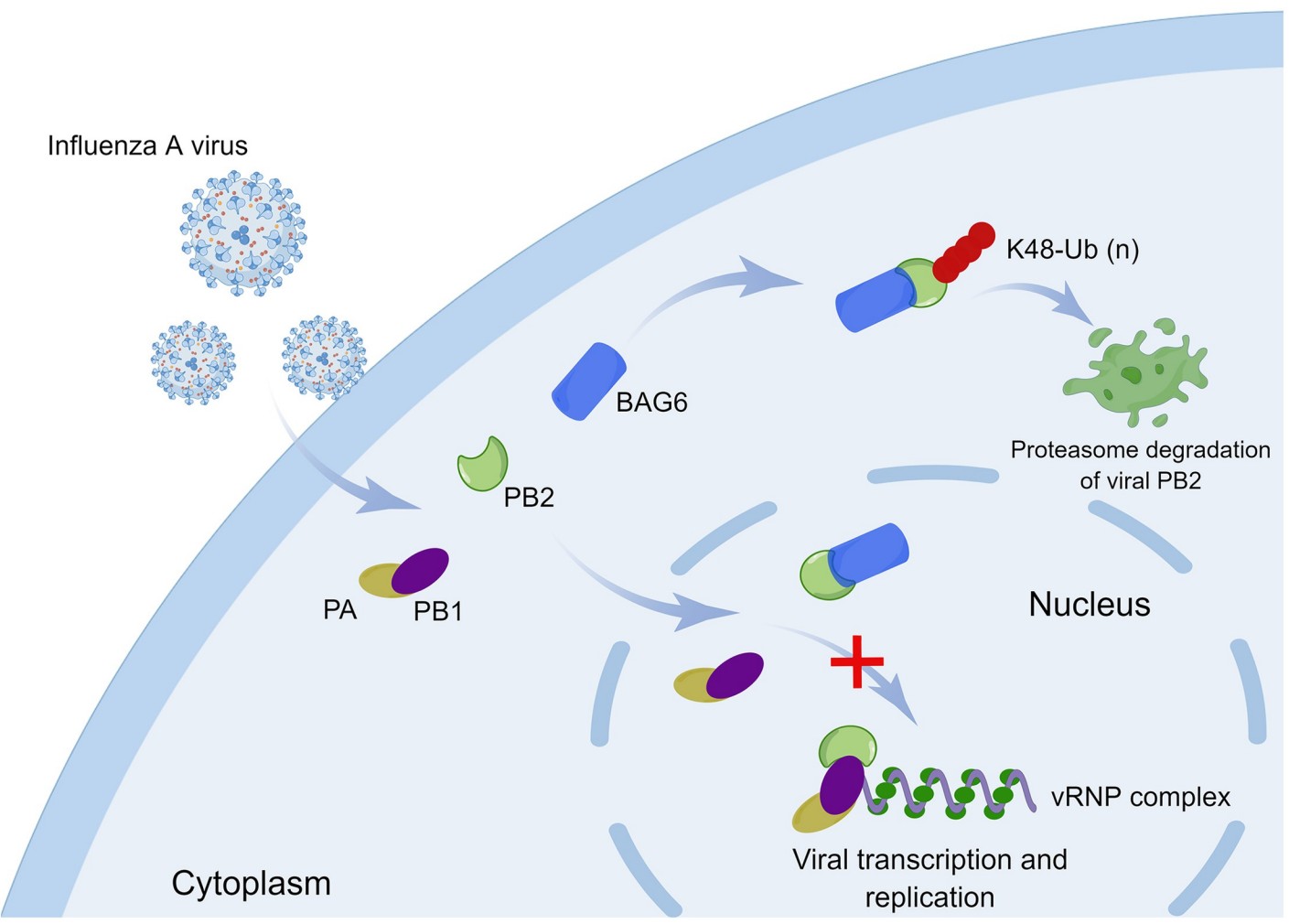

**Fig 10. Schematic diagram of a model for the mechanism by which BAG6 inhibits IAV replication.** During IAV infection, BAG6 directly interacts with the N-terminus of the IAV polymerase subunit PB2. The interaction between BAG6 and PB2 competes with PB1 for PB2 binding, which perturbs the formation of a functional RdRp complex, and further targets PB2 for K48-linked ubiquitination degradation, thereby inhibiting IAV replication (by Figdraw).

Antibodies used as follows: anti-BAG6 (B01P, Abnova, Taiwan, China), anti-PB2 (PA5-32220, Thermo Scientific, Massachusetts, USA), anti-PB1 (PA5-34914, Thermo Scientific, Massachusetts, USA), anti-β-actin (A1978, Sigma, Shanghai, China), anti-GAPDH (ab8245, Abcam, Shanghai, China), anti-PARP (9542, Cell Signaling Technology, Massachusetts, USA), anti-Flag (F1804, Sigma, Shanghai, China and 20543-1-AP, Proteintech, Chicago, USA), anti-HA (H9658, Sigma, Shanghai, China and 51064-2-AP, Proteintech, Chicago, USA), anti-RIG-I (20566-1-AP, Proteintech, Chicago, USA), anti-p-STAT1 (9167, Cell Signaling Technology, Massachusetts, USA), anti-ISG15 (sc166755, Santa Cruz, California, USA), anti-influenza A virus NP (A01506-40, GenScript, New Jersey, USA), Mouse monoclonal anti-influenza A virus M1 antibody (Lab Homemade, Beijing, China, ZL201910912806.0). Cycloheximide (CHX, HY-12320), MG132 (HY-13259C) and Chloroquine (CQ, HY-17589A) was purchased from MCE.

## Polymerase activity assay

A dual-luciferase reporter assay system (E1960, Promega, Wisconsin, USA) was used to compare the polymerase activities of different viral RNP complexes. PB2, PB1, PA, and NP gene segments of the indicated viruses (PR8, CN0606 and SD196) were separately cloned into the pCDNA3.1 expression plasmid. PB2, PB1, PA, and NP plasmids (125 ng each plasmid) along with the pLuci luciferase reporter plasmid (10 ng) and the renilla internal control plasmid (2.5 ng) were used to transfect cells. Cell cultures were incubated at 37°C. Cell lysates were analyzed at 24 h post-transfection for firefly and renilla luciferase activities using a GloMax 96 microplate luminometer (GM2000, Promega, Wisconsin, USA). Polymerase proteins were detected by Western blotting.

## BAG6 knockdown in mice by PPMO

Six-week-old BALB/c background female mice were purchased from Beijing Vital River Co. Ltd. (Beijing, China). Animal studies were carried out in specific pathogen-free barrier facilities. Facilities were maintained at an acceptable range of 20–26°C humidity of 30–70% on a 12-h dark/12-h light cycle. Peptide-conjugated PPMO were purchased from Gene Tools (USA). PPMO targeting mouse BAG6 gene (PPMO-BAG6) was designed as CTGGCACTAT-CACTCGGCTCCATG. A nontargeting PPMO control sequence (PPMO-NC) (CCTCTTACCTCAGTTACAATTTATA), was used as control. Six to eight weeks BALB/c female mice were inoculated intranasally with 100 μg of PPMOs in 50 μL PBS for 2 days continuously.

## Mice experiments

Mice were anesthetized and intranasally inoculated with PR8 virus at a median tissue culture infectious dose ($TCID_{50}$) of 100 in 50 μL of PBS. Body weight and survival were monitored daily for 14 days. Lung tissue lysates were generated by homogenizing snap-frozen lung tissues twice (20 s each time) in MEM medium and centrifuging the lung suspensions at 2000 rpm for 15 min. $TCID_{50}$ assays were performed using MDCK cells and $TCID_{50}$ was calculated as previously described [58]. A portion of the lung from each euthanized mouse at 3 and 5 dpi was fixed in 10% phosphate-buffered formalin, embedded in paraffin, sectioned and stained with hematoxylin and eosin (H&E). Immunohistochemical staining of viral antigen was performed as described previously [59].

## Quantitative real-Time PCR (qRT-PCR)

Total RNA from cells was extracted using an RNA isolation kit (Thermo Scientific, Massachusetts, USA). First-strand complementary DNA was synthesized from 1 μg of total RNA using a TransScript RT reagent kit (TransGen, Beijing, China), and oligo dT primer were used for detecting host genes. For the detection of influenza virus mRNA and vRNA, oligo dT primer and uni-12 primer (5′-AGCAAAAGCAGG-3′) [60] were respectively used to generate cDNAs. Generated cDNA was subjected to qPCR in a 25 μL reaction volume using 2X M5 HiPer SYBR Premix EsTaq (with Tli RNaseH) (Mei5bio) (S1 Table). Human β-actin genes were amplified for normalization of the cDNA amount used in qPCR. Reactions were conducted in triplicate, and the data were analyzed using the $2^{-\Delta\Delta Ct}$ method.

## Immunoprecipitation and western blotting

Cells were lysed in lysis buffer (50 mM Tris-Cl at pH 8.0, 150 mM NaCl, 1% Tri-ton X-100, 1 mM DTT, 1× complete protease inhibitor cocktail, and 10% glycerol) and pre-cleared with

Protein A/G PLUS-Agarose beads (sc-2003, Santa Cruz Biotechnology, USA) for 2 h at 4˚C. The lysates were then immunoprecipitated with indicated antibodies or isotype-matched control antibodies plus protein G Sepharose beads at 4˚C for 2–4 h. The beads were then washed three times and boiled. Protein samples were analyzed by Western blotting. For Western blotting, protein samples were mixed with 6× loading buffer supplemented with 10% β-mercaptoethanol, heated at 100˚C for 5 min and separated on a 10% SDS-PAGE under reducing conditions. After electrophoresis, protein samples were electroblotted onto polyvinylidene difluoride membranes (Bio-Rad, CA, USA) and blocked for 2 h in Tris-buffered saline (10 mM Tris-HCl, pH 8.0, containing 150 mM NaCl) containing 5% (w/v) non-fat dry milk and 0.5‰ (v/v) Tween-20. The blots were incubated with the primary antibodies overnight at 4˚C. The next day, the blots were incubated with corresponding horseradish peroxidase (HRP)-conjugated secondary antibodies for 1 h at room temperature (RT). HRP antibody binding was detected using a standard enhanced chemiluminescence (ECL) kit (wbluf0500, Millipore, USA).

## Protein degradation assay

HEK293T cell lines were cotransfected with PB2-Flag plasmid with empty vector or BAG6-HA plasmid. After 24 h post-transfection, CHX (50 μg/mL) was added to the medium and cells were harvested at different time points. Western blotting of cell lysates was performed to detect PB2-Flag, BAG6-HA and β-actin. To investigate the cellular pathway of BAG6-mediated PB2 degradation, at 24 h post-transfection, or infection with PR8 (H1N1), MG132 (20 μM) or CQ (150 μM) was added to the medium, and the cells were harvested 6 h after treatment. Three independent experiments were performed.

## Ubiquitination assay

HEK293T cells were transfected with HA-tagged ubiquitin (Ub-HA) plasmids, BAG6-HA or empty plasmids, and PB2-Flag or PB2-Flag with Lysine mutations. At 24 h post-transfection, the cells were treated with MG132 (20 μM) for 6 h and then lysed in 1% SDS lysis buffer. After being boiled for 10 min, the lysates were diluted 10 times with cold lysis buffer supplemented with 1× complete protease inhibitor cocktail, 10 mM DTT, and 10 mM NEM. PB2-Flag protein was then purified with anti-Flag antibody and subjected to western blotting analysis. Ubiquitinated PB2-Flag was detected by anti-HA antibody.

## Site statistics for amino acid residue 189 of the PB2 protein

We downloaded all available PB2 amino acid sequences of human and avian origin for influenza viruses of subtypes H1, H3, H5, H7, and H9 from the Global Initiative on Sharing Avian Influenza Data (GISAID, www.epicov.org) (date of access: Jan. 22, 2024), and multiple sequence alignment was performed using MAFFT v7.0 [61], and then the amino acid categories and percentages of the 189 amino acid residues were counted.

## Confocal microscopy

Cells on coverslips were washed twice with prewarmed PBS and fixed with 4% paraformaldehyde for 15 min at room temperature. Cells were subsequently permeabilized with immunostaining permeabilization buffer containing Triton X-100 (P0096, Beyotime, Shanghai, China) for 10 min and blocked with QuickBlock blocking buffer (P0220, Beyotime, Shanghai, China) for 20 min at room temperature. Fixed cells were incubated with indicated antibodies diluted in immunostaining primary antibody dilution buffer at 4˚C overnight. Coverslips were then

washed three times with PBS and incubated with Alexa Fluor 488-conjugated secondary antibodies or Alexa Fluor 555-conjugated secondary antibodies for 1 h at 37°C. Coverslips were finally washed three times and mounted onto microscope slides with DAPI staining solution (C1006, Beyotime, Shanghai, China) for 8 min and examined by confocal microscopy. Immunoassayed cells were visualized using a Nikon super-resolution laser scanning confocal microscope under a 100-time oil objective and analyzed using the Imaris 9.2 platform.

### Prediction of protein structure and complex

An online software, ColabFold, was used for the prediction of protein structures and complexes by combining the fast homology search of MMseqs2 with AlphaFold2 or RoseTTAFold. Sequence alignments/templates are generated through MMseqs2 and HHsearch [62]. The amino acid sequences of BAG6 protein N-terminal 1-255aa and PB2 protein N-terminal 1-247aa were used for the prediction of protein complex at the open-source website ColabFold v1.5.2 (https://colab.research.google.com/github/sokrypton/ColabFold/blob/main/AlphaFold2.ipynb#scrollTo=mbaIO9pWjaN0), and the predicted results were viewed and analyzed using PyMOL v2.5.0.

### Cell viability assay

WT A549 and BAG6-KO A549 cells or WT HeLa and BAG6-KO HeLa cells were plated in 96-well plates at a density of 2000 cells in 100 μL of medium per well. The cell viability was then assayed according to the CCK8 kit instructions (C0042, Beyotime, Shanghai, China).

### Apoptosis detection

Annexin V-PE/7-AAD Apoptosis Detection Kit (A213-01, Vazyme, Nanjing, China) was used for quantifying the cell apoptosis rates. The cells were collected by centrifugation and resuspended in 100 μL of 1× binding buffer supplemented with 5 μL of phycoerythrin (PE) annexin V and 5 μL of 7-aminoactinomycin D (7-AAD). Data were analyzed in FlowJo.

### Statistical analysis

For all the bar graphs, data are shown as the mean ± SD. All statistical analyses were performed using GraphPad Prism software v.8.00 (GraphPad Software). Unpaired Student's t-test was used to perform statistical analysis between the two groups. The Kaplan-Meier method was employed for survival analysis. Differences in means were considered statistically significant at $P < 0.05$; significance levels are as follow: $^*P < 0.05$; $^{**}P < 0.01$; $^{***}P < 0.001$; NS, not significant.

### Supporting information

**S1 Fig. BAG6 inhibits H1N1 replication in HeLa cells.** (A and B) HeLa cells were transfected with BAG6-HA or Empty vector (EV) plasmids. At 24 h post-transfection, the cells were infected with IAV H1N1 (MOI = 1.0). The viral NP expression was measured by western blotting at different time points postinfection, as indicated (A). Viral titers in the supernatants were determined by TCID$_{50}$ assay at 12, 24 or 36 h post-infection (B). (C and D) BAG6-KO and BAG6-WT HeLa cells were infected with IAV H1N1 (MOI = 1.0), and the NP and M1 expression in the cell lysates (C) and viral titers in the supernatants (D) were determined by western blotting and TCID$_{50}$ assay, respectively, at the indicated time points. Data presented as means ± SD and are representative of three independent experiments. $^*p < 0.05$,

$**p < 0.01$, $***p < 0.001$, Unpaired Student's t test.
(TIF)

**S2 Fig. BAG6 knockout did not affect cell growth viability.** Cell viability of BAG6-KO A549 and BAG6-WT A549 cells was measured using the CCK8 kit.
(TIF)

**S3 Fig. The expression of BAG6 can be suppressed by IAV infection.** (A-C) A549 cells were infected with IAV H1N1 (A) or H7N9 (B) or H9N2 (C) (MOI = 1.0), and the mRNA and protein expression of BAG6 were determined by quantitative real-time PCR (left panels) and western blotting (right panels), respectively, at the indicated time points postinfection. (D) A549 cells were infected with IAV H1N1 or H7N9 or H9N2 with different MOI as indicated. The expression of BAG6 protein was determined by western blotting. Data presented as means ± SD and are representative of three independent experiments. $*p < 0.05$, $**p < 0.01$, $***p < 0.001$, Unpaired Student's t test.
(TIF)

**S4 Fig. BAG6 does not affect IAV-induced apoptosis.** (A) BAG6-HA expression plasmid or empty vector was transfected into A549 cells for 24 h and then infected with PR8 virus at 1.0 MOI. The expression of PARP, cleaved PARP, and viral NP and M1 proteins at 0, 4, 8, 12, 24 and 36 h post-infection was detected using western blotting. β-actin detection was used as loading control. (B) A549 cells were untreated, transfected with BAG6 only, infected with IAV only, or both transfected with BAG6 and infected with IAV, and the cells were collected by centrifugation and were resuspended in 100 μL of 1x binding buffer supplemented with 5 μL of PE annexin V and 5 μL 7-AAD. Fluorescence of the stained cells was then analyzed using flow cytometry. Data presented as means ± SD and are representative of three independent experiments. $*p < 0.05$, $**p < 0.01$, $***p < 0.001$, Unpaired Student's t test.
(TIF)

**S1 Table. Amino acid residues at site 189 of the PB2 protein of different subtypes of influenza virus.**
(DOCX)

**S2 Table. List of primer pairs used in this study.**
(DOCX)

**S1 Data. The raw data of western blot and IF in this study.**
(PDF)

## Author Contributions

**Conceptualization:** Yong Zhou, Rui Zhang, Juan Pu.

**Data curation:** Yong Zhou, Tian Li, Rui Zhang.

**Formal analysis:** Yong Zhou, Tian Li, Nianzhi Zhang, Rui Zhang, Juan Pu.

**Funding acquisition:** Jinhua Liu, Rui Zhang, Juan Pu.

**Investigation:** Yong Zhou, Tian Li, Yuxin Guo, Xiaoyi Gao, Wenjing Peng, Sicheng Shu, Chuankuo Zhao.

**Methodology:** Yong Zhou, Tian Li, Rui Zhang, Juan Pu.

**Resources:** Yong Zhou, Tian Li, Yunfan Zhang, Di Cui.

**Visualization:** Yong Zhou, Rui Zhang.

**Writing – original draft:** Yong Zhou, Rui Zhang.

**Writing – review & editing:** Yong Zhou, Honglei Sun, Yipeng Sun, Jinhua Liu, Jun Tang, Rui Zhang, Juan Pu.

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
