## [Decision Letter · Decision Letter 0]

4 Dec 2023

Dear Dr. Zhang,

Thank you very much for submitting your manuscript "BAG6 inhibits influenza A virus replication by inducing viral polymerase subunit PB2 degradation and perturbing RdRp complex assembly" for consideration at PLOS Pathogens. As with all papers reviewed by the journal, your manuscript was reviewed by members of the editorial board and by several independent reviewers. In light of the reviews (below this email), we would like to invite the resubmission of a significantly-revised version that takes into account the reviewers' comments.

The reviewers appreciated the attention to an interesting topic but had mixed views on the extent to which the conclusions are supported by the data. Reviewer 2 raised important considerations related to already described functions of BAG6, which could account for some of the observed effects on IAV replication via a distinct mechanism to that described. Thus, in revising the manuscript, it is important to assess the impact of BAG6 overexpression and knockout on innate antiviral responses. Revisions should also focus on increasing the rigor of the experiments through the inclusion of appropriate controls, as suggested by all three reviewers.

We cannot make any decision about publication until we have seen the revised manuscript and your response to the reviewers' comments. Your revised manuscript is also likely to be sent to reviewers for further evaluation.

Sincerely,

Anice C. Lowen

Academic Editor

PLOS Pathogens

Benhur Lee

Section Editor

PLOS Pathogens

Kasturi Haldar

Editor-in-Chief

PLOS Pathogens

orcid.org/0000-0001-5065-158X

Michael Malim

Editor-in-Chief

PLOS Pathogens

orcid.org/0000-0002-7699-2064

The reviewers appreciated the attention to an interesting topic but had mixed views on the extent to which the conclusions are supported by the data. Reviewer 2 raised important considerations related to already described functions of BAG6, which could account for some of the observed effects on IAV replication via a distinct mechanism to that described. Thus, in revising the manuscript, it is important to assess the impact of BAG6 overexpression and knockout on innate antiviral responses. Revisions should also focus on increasing the rigor of the experiments through the inclusion of appropriate controls, as suggested by all three reviewers.

Reviewer's Responses to Questions

**Part I - Summary**

Reviewer #1: Virus-host interaction is particularly important for the regulation of influenza A virus (IAV) replication and pathogenesis. In the present manuscript, the authors found that overexpression of BAG6 inhibited IAV replication, whereas knockout of BAG6 enhanced IAV replication. By performing luciferase reporter assay, the authors found that BAG6 inhibited the polymerase activity of IAV. Mechanistically, they found that BAG6 suppressed the RdRp activity of IAV through catalyzing the K48-linked ubiquitination and degradation of viral PB2 protein. Meanwhile, the ubiquitin-like (UBL) domain (17-92aa) and PB2-binding domain (124-186aa) of BAG6 were both required for mediating the PB2 degradation and the impairment of virus replication. Notably, BAG6 had obvious antiviral effect in mice. The manuscript can be further improved by addressing the comments below.

Reviewer #2: This is an interesting work revealing the role of BAG6 (BCL2-associated athanogene 6) in restricting Influenza A virus (IAV) replication. The IAV is a major public health concern worldwide. During infection, the viral genome is replicated and translated by the RDRP complex, which consists of PB2, PB1, PA, NP and the vRNA. These polymerases are relatively conserved across the IAV subtypes and are involved in host tropism and host adaptation. In Fig.1, both for the avian and human IAV, overexpression of BAG6 reduces viral translation and virus titer. Simultaneously, in Fig.2, BAG6 KO significantly enhanced virus replication. During this course, the authors found a decrease in BAG6 transcription and translation, and is evident that BAG6 is being downregulated. The authors propose a mechanism of antiviral activity in which the N-terminal region of BAG6 interacts with PB2 to inhibit replication and by targeting PB2 for degradation.

The manuscript is interesting and could be of interest to the field, however, there is already reports in the literature that suggests BAG6 degrades the protein aggregates by K48-linked ubiquitination (PMID: 23275523), and additionally, during RNA virus infection, it negatively regulates RLR-mediated antiviral response by inhibiting the aggregation/ complex formation of the multi-subunit protein complex, virus-induced-signaling adapter (VISA) in the RIG-I pathway, by K48-linked ubiquitination (PMID: 36045679). These manuscripts are not considered and the effects on the innate response are not addressed. Cellular factors might be contributing to the down-regulation of BAG6 so that RIG-I pathway, and this is not ruled out. It is essential that the authors connect if any of the viral proteins are involved in the down-regulation of BAG6 transcription. Mechanistically, the manuscript does not go deep enough to advance the knowledge from previous reports

Reviewer #3: The manuscript by Zhou et al describes the role of BAG6 (BCL2-associated athanogene 6) as restriction factor in influenza virus infection. Overexpression of BAG6 leads to about a log drop in viral titer, while knockout of BAG6 increase viral titers by about a log in A549 and Hela cells. The authors used morpholinos to knock down BAG6 expression in mice, and found that this led to increased viral titers, weight loss and death, whereas no effects were seen in the PBS-infected BAG6 knock down mice or the infected wt mice. Finally, the authors used pull-down assays to demonstrate that BAG6 binds to the IAV RNA polymerase subunit PB2 and that it can induce the degradation of PB2. The authors propose that BAG6 interacts with the N-terminus of PB2 and interferes with RNA polymerase assembly or induces PB2 degradation. Generally, the manuscript is well-organized, but there are many grammatical issues (e.g. line 845 “A549 cells was transfected”), and the data appear to support the conclusions. There are a couple of points that would need attention.

**Part II – Major Issues: Key Experiments Required for Acceptance**

Reviewer #1: 1. Lines 135-136: BAG6 plays an important role in inhibiting the replication of different influenza virus subtypes from avian and human. The authors displayed the anti-viral effect of BAG6 on the replication of H1N1, H7N9, and H9N2 viruses. It will be more informative to determine whether BAG6 also inhibits the replication of highly pathogenic H5N1 virus.

2. In Fig 5B, the expression level of viral M1 protein was very low, and the size of extra nonspecific band was nearly overlapped with that of the viral M1 protein. So it's almost impossible to identify where the actual band of M1 protein is localized?

3. In Fig 6, BAG6 induces viral PB2 degradation through K48-linked ubiquitination. It will be highly desired to determine which amino acid residue of PB2 is modified with K48-linked ubiquitination. In such case, whether the mutation of the key amino acid residue in PB2 can prevent PB2 from being degraded by BAG6?

Reviewer #2: Major issues:

1) The role of type-I IFNs and innate cytokines are not considered, especially in light of previous publications. The mechanism is not studied in depth and the conclusions may not be supported without further investigation.

2) Figure 1. overexpression of BAG6 may result in immune activation in A549 cells. Type-I IFNs needs to be measured.

3) Figure 2, the same as in point 2 above. What are the effects of knockout in cytokines, IFNs and ISGs?

4) Figure 3. PPMOs could be also inducing immune activation. Controls need to include measuring IFNs and ISGs.

5) Figure 4 needs to show control expression of all transfected proteins

6) Figure 5, all IPs are done using wrong controls. Figure 5A, the proper control for the IP is the absence of BAG6-HA in the presence of PR8 infection. This is to ensure that there is no unspecific binding of the viral proteins to the beads. The same in Figure 5B, a control is needed without BAG6-HA in the presence of overexpressed Flaf proteins (at least PB2).

7) Figure 5C lacks controls, each protein needs to be overexpressed separately. Also, additional panels with more cells need to be shown.

8) Figure 6C and D need to be compared to cells without MG132 as control. Also these two experiments need to be repeated without HA-Ub overexpression and instead immunoblotting for endogenous ubiquitin with anti-Ub specific antibodies.

9) Figure 7D needs controls with individual proteins.

10) Fig.6, it is essential to show the rescue of PB2 degradation under MG132 treatment in infection conditions. Also, check the PB1 and NP levels under MG132 treatment since it was found to pulled down with BAG6 in Fig. 5A.

11) Fig.7, It will be appropriate to verify the effect of N-terminus BAG6 (1-225 aa) in the BAG6 KO cells under H1N1 infection.

Reviewer #3: (No Response)

**Part III – Minor Issues: Editorial and Data Presentation Modifications**

Reviewer #1: 1. Lines 113-114: "Consistently, the ectopic expression of BAG6 also caused a significant reduction in viral NP protein expression (Fig 1C) and viral loads (Fig 1D) of H7N9 and H9N2." The authors described the role of BAG6 in virus replication as "significant". The authors had better specifically provide the magnitude of the changes in virus titer. This lack of proper quantitative description of the data obtained is further complicated by the lack of specific numeric data in the figures presented, which makes it very difficult for the reader to estimate the magnitude of the changes observed.

2. In all Figures, the abbreviation format needs to be uniform, such as dpi and d.p.i. in Fig 3.

3. Line 172, BAG-KO should be BAG6-KO.

Reviewer #2: Minor issues:

1) Figure 3 also needs additional controls with PPMOs but without infection. The same for the IHCs in E and F, needs controls without infection.

2) In Fig.1 the authors can also check the densitometry of NP/M in the western blot to see if any correlation appears between the IAV subtypes in the presence/absence of BAG6 overexpression.

3) The authors can show the expression of RIG-I in the blot (Fig. 2A, BAG6-wt and KO and H1N1 infection).

4) For H1N1 infection in the PBS, PPMO-NC, and PPMO-BAG6 treated groups (Fig. 3, panel E), we see an appreciable and statistically significant viral titer on day 3 and as expected, it goes down on day 5. It will be informative if the authors can show the expression level of BAG6 on days 3 and -5 in these tissue samples.

5) Check the nucleotide sequence of Uni-12 Primer and highlight the source/ reference.

6) Fig.5 Panel A, under PR8 infection, BAG6 interacts with the PB2, PB1 and NP. PB2 and NP are pulled down, almost in equal amounts, with BAG6. How do you explain the discrepancy with Fig.5B (where only PB2 interacts with BAG6)? It could also be possible that the association of polymerase complex (PB2, PB1, PA and NP) on the vRNA is sensing the BAG6 and hence you get an interaction in Fig.5A.

7) Lines 139-140: BAG6-deficient mice were generated using a peptide-conjugated phosphorodiamidate morpholino oligomers (PPMOs)”. This sentence is incorrect. These are not deficient mice, this is a knockdown. Please correct text.

8) Lines 189-190:. The M1-Flag was used as a negative control. The results showed that BAG6-HA was only able to directly interact with PB2-Flag (Fig 5B, lane 3)…” . these experiemnst don’t whow direct interactions. Purified proteins would be needed in in vitro binding assays. Please remove ‘directly’ and discuss the potential for indirect interactions.

Reviewer #3: 1. Line 150: please clarify “extremely higher”

2. Figure 7B/C: The figure legend appears to be incorrect as it refers to the IP flag as Fig. 7C. This should be 7B, and the HA IP should be 7C.

3. Figure 7C: the IP has two Flag blots. Should the bottom one be part of the input?

4. In Fig. 7D, the 1:255-HA constructs keeps PB2 in the cytoplasm, suggesting that it affects PB2 import into the nucleus and the functioning of the PB2 NLS (which resides near the C-terminus of PB2). The authors only consider the BAG6 NLS in their discussion. In addition, it would be useful if the authors added controls to show/confirm the localization of PB2 and BAG6 alone.

5. Line 242. “Targeting for subcellcular localization” appears to be missing a word (e.g., cytoplasmic).

6. The authors claim that BAG6 binds to the PB2 N-terminus, but they provide no PB2 mutational data (e.g. PB2 truncations) to support this. The language should be adjusted.

PLOS authors have the option to publish the peer review history of their article (what does this mean?). If published, this will include your full peer review and any attached files.

Reviewer #1: No

Reviewer #2: No

Reviewer #3: No
---

## [Decision Letter · Decision Letter 1]

9 Mar 2024

Dear Dr. Zhang,

We are pleased to inform you that your manuscript 'BAG6 inhibits influenza A virus replication by inducing viral polymerase subunit PB2 degradation and perturbing RdRp complex assembly' has been provisionally accepted for publication in PLOS Pathogens.

Best regards,

Anice C. Lowen

Academic Editor

PLOS Pathogens

Benhur Lee

Section Editor

PLOS Pathogens

Michael Malim

Editor-in-Chief

PLOS Pathogens

orcid.org/0000-0002-7699-2064

Reviewer Comments (if any, and for reference):

Reviewer's Responses to Questions

**Part I - Summary**

Reviewer #1: In this study, the authors identified BAG6 as a host restricting factor for the replication of IAV. Detailed investigation revealed that BAG6 mediates the K48-linked polyubiquitination and proteasomal degradation of viral PB2 protein, thereby inhibiting the activity of the viral RNA-dependent RNA polymerase. In the revised version of the manuscript, the authors further found that the lysine residue at position 186 of PB2 is the targeting site of BAG6-mediated polyubiquitination. Overall, this study is well performed and the findings are interesting.

Reviewer #2: In this revision the authors perform experiments to answer most of the previous weaknesses raised, however there are still issue that need to be addressed:

Reviewer #3: The authors have done additional controls and significantly improved the manuscript. I support publication.

**Part II – Major Issues: Key Experiments Required for Acceptance**

Reviewer #1: My major comments for the initial version of the manuscript have been addressed. I have no further issues to point out at this stage.

Reviewer #2: Major issues:

1. Figure 7C and D need to be compared to cells without MG132 as a control. The authors did not address this issue claiming that the levels of ubiquitination are barely detectable without MG132 treatment, because the protein substrates are quickly degradated after ubiquitination modification. Whether this is true or not, the authors need to show this as control. This is exactly the point, that the authors show what happens between +/- MG132, and can demonstrate the levels of ubiquitinated proteins. This can be done with overexpressed proteins.

2. The second point is about detecting endogenous ubiquitin (without the use of overexpressed HA-Ub). In this case the author’s response was that ‘it is difficult to monitor endogenous ubiquitin due to its relatively low quantity’. This statement is simply incorrect. It is simply incorrect that endogenous ubiquitin in the cell is present in low levels. These experiments are standard in the ubiquitin field. Endogenous ubiquitin is present in highly enough levels to be able to be detected with anti-ubiquitin antibodies by immunoblot, and a simple immunoprecipitation of PB2 (overexpressed, or even better during infection), if indeed ubiquitinated, should be able to detect ubiquitinated PB2. Actually, this experiment can be done with and without MG132, if the authors are correct that ubiquitinated PB2 is degraded quickly.

Reviewer #3: N/A

**Part III – Minor Issues: Editorial and Data Presentation Modifications**

Reviewer #1: I have no further minor comments for the revised version of the manuscript.

Reviewer #2: 1. Please look carefully at labeling. In Fig. 4C it says P-SAT1. correct the spelling to P-STAT1.

2. Line 235-237: “…BAG6 could co-localize with PB2 and promote its localization in both nuclei and cytoplasm of cells (Fig 6E)…”

Is the opinion of this reviewer that the re-localization of PB2 ‘promoted’ by BAG6 is not clear. Actually the colocalization of PB2 with BAG6 is also not very clear. They are both in the nucleus, but the overlap is not clear. If the authors want to make this statement about re-localization, then this needs to be demonstrated with a nucleo-cytoplasmic fractionation. Otherwise remove the claim.

3. The authors showed the rescue of PB2 degradation by MG132 treatment under the H1N1 (PR8) infection (Fig. 7C). I seems that from Figure 6A, there are also interactions with NP and. It would be appreciated if the authors also show the NP blot in Fig. 7C. This could open a new direction that BAG-6 interacts both with PB1, PA and NP, however, it seems that is selectively targets PB2 for degradation? Can the authors show the blot for NP in Figure 7C?

Reviewer #3: N/A

PLOS authors have the option to publish the peer review history of their article (what does this mean?). If published, this will include your full peer review and any attached files.

Reviewer #1: **Yes: **Chengjun Li

Reviewer #2: No

Reviewer #3: No

---

## [Editor Report · Acceptance letter]

13 Mar 2024

Dear Dr. Zhang,

We are delighted to inform you that your manuscript, "BAG6 inhibits influenza A virus replication by inducing viral polymerase subunit PB2 degradation and perturbing RdRp complex assembly," has been formally accepted for publication in PLOS Pathogens.

Best regards,

Michael Malim

Editor-in-Chief

PLOS Pathogens

orcid.org/0000-0002-7699-2064